# Transcriptomics-Based Phenotypic Screening Supports Drug Discovery in Human Glioblastoma Cells

**DOI:** 10.3390/cancers13153780

**Published:** 2021-07-27

**Authors:** Vladimir Shapovalov, Liliya Kopanitsa, Lavinia-Lorena Pruteanu, Graham Ladds, David S. Bailey

**Affiliations:** 1IOTA Pharmaceuticals Ltd., St Johns Innovation Centre, Cowley Road, Cambridge CB4 0WS, UK; vladimir.shapovalov@iotapharma.com (V.S.); liliya.kopanitsa@iotapharma.com (L.K.); lavinia.pruteanu@iotapharma.com (L.-L.P.); 2Department of Pharmacology, University of Cambridge, Tennis Court Road, Cambridge CB2 1PD, UK; grl30@cam.ac.uk

**Keywords:** glioblastoma, drug-inducible gene expression, Mardepodect, Regorafenib, drug targets, tumor antigens

## Abstract

**Simple Summary:**

Glioblastoma (GBM) remains a particularly challenging cancer, with an aggressive phenotype and few promising treatment options. Future therapy will rely heavily on diagnosing and targeting aggressive GBM cellular phenotypes, both before and after drug treatment, as part of personalized therapy programs. Here, we use a genome-wide drug-induced gene expression (DIGEX) approach to define the cellular drug response phenotypes associated with two clinical drug candidates, the phosphodiesterase 10A inhibitor Mardepodect and the multi-kinase inhibitor Regorafenib. We identify genes encoding specific drug targets, some of which we validate as effective antiproliferative agents and combination therapies in human GBM cell models, including HMGCoA reductase (*HMGCR*), salt-inducible kinase 1 (*SIK1*), bradykinin receptor subtype B2 (*BDKRB2*), and Janus kinase isoform 2 (*JAK2*). Individual, personalized treatments will be essential if we are to address and overcome the pharmacological plasticity that GBM exhibits, and DIGEX will play a central role in validating future drugs, diagnostics, and possibly vaccine candidates for this challenging cancer.

**Abstract:**

We have used three established human glioblastoma (GBM) cell lines—U87MG, A172, and T98G—as cellular systems to examine the plasticity of the drug-induced GBM cell phenotype, focusing on two clinical drugs, the phosphodiesterase PDE10A inhibitor Mardepodect and the multi-kinase inhibitor Regorafenib, using genome-wide drug-induced gene expression (DIGEX) to examine the drug response. Both drugs upregulate genes encoding specific growth factors, transcription factors, cellular signaling molecules, and cell surface proteins, while downregulating a broad range of targetable cell cycle and apoptosis-associated genes. A few upregulated genes encode therapeutic targets already addressed by FDA approved drugs, but the majority encode targets for which there are no approved drugs. Amongst the latter, we identify many novel druggable targets that could qualify for chemistry-led drug discovery campaigns. We also observe several highly upregulated transmembrane proteins suitable for combined drug, immunotherapy, and RNA vaccine approaches. DIGEX is a powerful way of visualizing the complex drug response networks emerging during GBM drug treatment, defining a phenotypic landscape which offers many new diagnostic and therapeutic opportunities. Nevertheless, the extreme heterogeneity we observe within drug-treated cells using this technique suggests that effective pan-GBM drug treatment will remain a significant challenge for many years to come.

## 1. Introduction

Glioblastoma (GBM) is characterized by pronounced cellular heterogeneity, with different glioblastoma cell lineages presumed to emanate from glioma stem cells (GSCs) within the same patient tumor [1]. GSCs often retain neural differentiation characteristics, although they do not terminally differentiate [2]. Transcriptomics studies have previously identified gene expression signatures that correlate with patient survival [3]. Using single-cell RNA sequencing, individual tumor cells can be positioned within a spectrum spanning proneural to mesenchymal cell types, with the mesenchymal phenotype correlating with significantly poorer patient survival [4]. Moreover, tumor-initiation studies with cell surface marker-enriched GBM populations, xenografted into immunodeficient mouse models, show that these cells retain their capability to re-form the full spectrum of proneural to mesenchymal phenotypes observed in the original patient tumors [5], emphasizing the phenotypic plasticity and stem-like characteristics of GBM tumor cells.

At a genetic level, attention has centered on the ‘driver’ mutations implicated in the development of GBM and other cancers, confirmed by sequencing at the single-cell level [6]. Nevertheless, despite an increasing understanding of the molecular evolution of such tumors, and the development of powerful new approaches such as immunotherapy to target them, GBM clinical outcomes remain poor [7]. New drug and vaccine targets which translate into effective therapies are urgently required.

One of the hallmarks of GBM is its extreme resistance to growth inhibition by traditional anti-proliferative drugs as monotherapies, such as EGFR inhibitors [8]. To address this challenge, several novel GBM treatment modalities, such as combination drug therapy [9], immunotherapy [10], and adjuvant-enabled CAR-T cell therapy [11] are being developed. Work in other cancers has highlighted the importance of phenotypic plasticity in cancer initiation, progression, and resistance to therapy [12], and progress in GBM treatment is likely to mirror that in other cancers, such as multiple myeloma and melanoma where phenotyping is central to therapy selection [13,14].

The last decade has seen a resurgence in phenotypic screening, largely due to the realization that sifting through the thousands of potential therapeutic targets delivered from genomics, one by one, is both time-consuming and expensive [15,16]. Building oncology drug discovery campaigns on mechanistically validated chemical compounds and relevant phenotypic screens is now an established route to accelerated drug discovery [17,18]. Moreover, rapidly repositioning existing drugs for use in GBM could provide radically new and effective GBM pipelines [19].

Drug-focused chemical biology has one further big advantage—it provides insights into the quality of drug candidates on their way to drug development. The genome-wide drug induced gene expression (DIGEX) techniques employed here provide a formidable platform for comparing drug action, using thousands of ‘reporter’ gene expression data points to tease apart drug properties. The approach can also provide new insights into the dynamics of the drug response, a feature that could prove invaluable in an adaptive clinical trial setting.

Previous work from our laboratory has used genome-wide DIGEX to define the GBM cell phenotype and its modulation by drugs. In that work, we focused on the prototypic PI3K growth inhibitor LY-294002 and the natural product Fucoxanthin as chemical probes [20]. In the current study, we extend these detailed observations to a suite of three well-characterized GBM cell lines—U87MG, A172, and T98G—and two further growth inhibitory drugs in the clinic, the phosphodiesterase 10A (PDE10A) inhibitor Mardepodect (PF-02545920) and the multi-protein kinase inhibitor Regorafenib (Stivarga, BAY 73-4506).

Mardepodect is a CNS penetrant PDE10A inhibitor [21,22], developed by Pfizer initially for schizophrenia [23] and later repositioned for Huntington’s Chorea within the AMARYLLIS clinical trial [24]. Mardepodect is thought to increase cAMP/PKA signaling in medium spiny neurons of the human striatum, which in turn leads to potentiation of dopamine D1 receptor signaling with concomitant inhibition of dopamine D2 receptor signaling. However, although safe and well tolerated, and capable of crossing the blood–brain barrier, Mardepodect failed to achieve satisfactory therapeutic endpoints in either schizophrenia or Huntington’s chorea. Here, we show that Mardepodect potently inhibits the growth and proliferation of GBM cells, raising the new possibility of its repositioning in GBM.

Regorafenib was originally approved for patients with treatment-refractory metastatic colorectal cancer as an adjunct to sorafenib treatment [25]. Regorafenib has a radically different molecular mode of action to that of Mardepodect, promiscuously targeting many protein kinases including VEGFR-1, -2, -3, TIE 2, PDGFR, FGFR, KIT, RAF-1, RET, and BRAF [26]. Regorafenib has already been evaluated for its effects on GBM within the REGOMA clinical trial [27] and is currently a component of the ongoing GBM AGILE adaptive clinical trial [28].

Although to date no drug, including Regorafenib, has provided effective therapy for GBM, it is still important to define the GBM cell response to every FDA approved drug showing promise in GBM, since such drugs are valuable assets and may elicit responses that can be exploited in new ways, perhaps in combination therapy or phenotypic modulation.

Thus, in this study, we reposition the schizophrenia drug Mardepodect as a possible antiproliferative candidate in GBM. Using DIGEX we compare the effects of Mardepodect to those of Regorafenib, a drug already in clinical trials for GBM.

We chose to study Mardepodect and Regorafenib not only because the two drugs are clinical candidates, but also because they cover highly complementary pharmacological space and, in combination, might synergize in providing a novel way to address the pronounced drug resistance which characterizes GBM.

Previously, we have used a range of pharmacological probes, including PDE inhibitors, to show that raised cAMP levels are associated with growth inhibition in rat C6 glioma cells [29]. Here, we focus on human PDE10A, a dual specificity cyclic nucleotide phosphodiesterase that is expressed in GBM but has not previously been studied as a potential therapeutic target. Using DIGEX, we compare and contrast the transcription phenotypes accompanying growth inhibition by the PDE inhibitor Mardepodect with those of the kinase inhibitor Regorafenib, reasoning that this information might enable us to design new combination therapies targeting these two anti-proliferative signaling pathways.

## 2. Materials and Methods

### 2.1. Cells Used in This Study

Cell proliferation experiments were carried out in the well characterized established GBM cell lines U87MG, A172, and T98G. The human glioblastoma astrocytoma cell lines U87MG (ECACC 89081402) and A172 (ECACC 88062428) were obtained from the European Collection of Authenticated Cell Cultures. The T98G cell line was obtained from the American Type Culture Collection (ATCC^®^ CRL1690™, Manassas, VA, USA). The mutational landscapes of all three cell lines have been archived within Expasy (www.expasy.org/cellosaurus accessed on 15 November 2020) as U-87MG ATCC (RRID:CVCL_0022), A-172 (RRID:CVCL_0131), and T98G (RRID:CVCL_0556).

All cell lines were maintained in Dulbecco’s modified Eagle’s medium: Nutrient Mixture F-12 (DMEM/F12, Gibco, ThermoFisher, Loughborough, UK) supplemented with 10% fetal bovine serum (FBS, Sigma, Dorset, UK) and 5% antibiotic antimycotic solution (10,000 units of penicillin, 10 mg streptomycin, and 25 μg/mL amphotericin B, Sigma, Dorset, UK) at 37 °C in humidified atmospheres of 95% air and 5% CO_2_.

### 2.2. Compounds Used in This Study

The compounds LY-294002, Regorafenib, Mardepodect (PF-02545920), Atorvastatin, and Simvastatin were purchased from Sigma UK. AZD1480 and Ruxolitinib were purchased from Selleckchem (München, Germany). HG-9-91-01 and Icatibant were purchased from MedChemExpress (Insight Biotechnology Limited, Wembley, Middlesex, UK). WH-4-023 and WIN 64338 were purchased from Tocris (Bio-techne Ltd., Abingdon, UK). Stock solutions of all compounds, except Icatibant, were prepared in dimethyl sulfoxide (DMSO) before addition to culture medium for testing. Stock solutions of Icatibant were prepared in water.

### 2.3. Proliferation Assay

Inhibition of proliferation by compounds in the three glioblastoma cell lines was determined using the Cell Counting Kit-8 (CCK-8) assay (Sigma, UK), as described previously [20].

### 2.4. Drug Combination Assays and Their Analysis

To study the effects of combined treatments with Mardepodect, Regorafenib, LY-294002, and Fucoxanthin with inhibitors of JAK2 kinase, SIK1, and HMGCoA reductase, proliferation assays were performed, followed by an analysis of the observed combination effects with the additive Loewe synergy effect as a baseline model, using Combenefit software (version 2.021) [30] for analysis.

### 2.5. Microarray Analysis

Cells were seeded into T25 flasks at a density of 500,000 cells/flask and allowed to adhere and grow for 24 h. The culture medium was removed, and fresh medium containing test compound in 1% DMSO at the previously determined 72 h IC50 concentration was added to each flask. Control cells were treated with medium containing 1% DMSO alone. All experiments were performed in triplicate. The cells were visualized during culture using the EVOS Cell Imaging System (Thermo Fisher Scientific, UK).

After 24 h of treatment, the cells were trypsinized and total RNA was isolated using the RNeasy Mini kit (Qiagen, Manchester, UK) as described previously [20]. Expression analysis was performed on a Clariom S Human Array (Thermo Fisher Scientific, catalog number 902926) using a fixed number of probes per transcript and probe sets comprising a subset of 10 probes per gene, yielding >20,000 annotated genes, as documented by the NetAffx Analysis Center (www.affymetrix.com/analysis/netaffx/, accessed on 3 November 2020).

The raw data from all samples, in triplicate, were normalized taking average signal intensities, and an expression matrix was created by applying the Robust Multi-array Average (RMA) algorithm as a multi-chip model [31]. The control housekeeping gene intron/exon separation area under the receiving operating curve value threshold was selected as 0.8, ensuring high quality in all samples. Finally, the Clariom S chip probe sets were mapped to their Entrez IDs, resulting in a list of 18,316 identifiable protein-coding genes after exclusion of duplicate and non-coding gene sequence signals.

Specific genes were analyzed and annotated using the UniProt (www.uniprot.org accessed on 4 November 2020), Entrez (www.ncbi.nlm.nih.gov/gene accessed on 4 November 2020), and Gene Ontology (www.geneontology.org/ accessed on 5 November 2020) databases, together allowing identification of the putative function of particular genes, as well as the pathways in which they have been observed previously. In the analyses reported, UniProt protein entries are denoted in block capitals with NCBI Gene entries in italics. The Pharos database (www.pharos.nih.gov accessed on 9 June 2020) was used to identify potential drug targets based on their inherent druggability [32]. Color coding of these genes in the accompanying tables is based on the system used by the University of New Mexico (http://juniper.health.unm.edu/tcrd/ accessed on 9 June 2020). Principle component analysis (PCA) was used to study the reproducibility of gene expression among the different drug treatments. Reactome pathway analysis (https://www.reactome.org accessed on 3 June 2020) and gene network analysis (www.genenetwork.nl accessed on 4 January 2021) were also conducted, focusing on the 200 genes with most elevated, or lowered gene expression levels, and their subsets.

## 3. Results

### 3.1. Established Cell Lines Used in These Studies

Established cell lines, while possessing lengthy passage histories, provide well characterized, robust and relatively reproducible systems in which to compare drug responses, and studies with them have provided the bulk of the information we have on drug response in GBM. Well-adapted to large-scale tissue culture, established cell lines are also good starting points for dissecting the underlying biochemical and pharmacological processes governing the GBM drug response—and purifying the effectors involved. In this study, we have used the established cell lines U87MG, A172, and T98G to investigate drug-induced gene expression changes. All three cell lines have been completely sequenced and their mutational landscapes defined. They are also the three most highly represented GBM cell lines in more than 1000 studies reported in the GBM Drug Bank [33].

### 3.2. Compounds Used to Probe Drug-Induced Gene Expression

The primary focus of the current studies was to define the effects of the drug Mardepodect on GBM cell transcription. Mardepodect, a Phase 3 clinical candidate developed by Pfizer as PF-02545920 for schizophrenia and more recently repositioned for Huntington’s Disease, is a potent PDE10A inhibitor with CNS penetrant properties that may make it suitable for repositioning in GBM.

A second objective was to compare the Mardepodect response to that of the Bayer drug Regorafenib, approved by the FDA for colorectal carcinoma and currently in clinical trials for GBM [27,28]. Regorafenib is a well characterized multi-kinase inhibitor [26].

In an earlier study [20], we characterized the DIGEX profiles of two additional growth inhibitors, the chemical probe LY-294002 (another well characterized multi-kinase inhibitor), and Fucoxanthin (a xanthophyll natural product). Here, we use the profiles of both LY-294002 and Fucoxanthin as benchmarks against which to compare the gene expression profiles of Mardepodect and Regorafenib.

### 3.3. Growth Inhibition Characteristics of the Compounds

Dose response relationships are shown for all four compounds—Mardepodect, Regorafenib, LY-294002, and Fucoxanthin—in a standardized 72 h proliferation assay, using the three human GBM cell lines—U87MG, T98G and A172—growing in serum-containing medium (Figure 1).

The IC50_72h_ determined for Mardepodect varied from 32 µM for U87MG cells, to 5 µM for A172 cells, with T98G cells showing an intermediate IC50_72h_ of approximately 16 µM. In contrast, the dose–response relationships observed for Regorafenib and Fucoxanthin were similar in all three cell lines, giving an IC50_72h_ of approximately 10 µM.

### 3.4. DIGEX Profiles for the Four Treatments

Having determined the cellular IC50_72h_ for Regorafenib, Mardepodect, LY-294002 and Fucoxanthin, we prepared RNA samples from batches of cells treated for 24 h with these compounds at their IC50_72h_ concentrations, a standardized treatment protocol designed to capture significant DIGEX information under conditions of minimal toxicity. Twenty-four-hour dosing also mirrors a preferred clinical dosing regimen. The protocol gives highly reproducible DIGEX results, validated by PCA analysis (Appendix A).

### 3.5. Upregulated Genes Accompanying Drug Treatments in U87MG Cells

In initial experiments, we focused on the human glioblastoma cell line U87MG. Treatment of U87MG cells under standardized conditions with any of the four proliferation inhibitors—Mardepodect, Regorafenib, LY-294002, or Fucoxanthin—upregulated many genes when compared to control cells grown under the same culture conditions but without inhibitors. The 200 genes with the most elevated expression levels in each drug treatment were identified and compared in a four-way Venn diagram (Figure 2). Amongst the 200 gene sets in U87MG cells, the genes partitioned between different drug treatments were identified (Table 1).

### 3.6. Interpretation of the U87MG DIGEX Signatures

All four compounds produced richly complex DIGEX signatures in transcriptomic analyses of U87MG cells. However, relatively few genes were shared between treatments within the Top 200 upregulated genes (Figure 2 and Table 1). Concomitantly, many elevated genes were seen that were restricted to each treatment. These are documented in the Appendix A (Appendix A). From these initial results we concluded that:The DIGEX signature for each cell/treatment combination is reproducible and specific, confirmed by the PCA analysis (Appendix A).A GBM cell line such as U87MG, can radically change its DIGEX response when encountering different drug treatments, exhibiting considerable transcriptional plasticity.

We found the extreme diversity of gene expression produced within a single cell line by a single biological event—growth inhibition—somewhat surprising. However, the data show clearly that the U87MG cell line can deploy a vast range of specific drug-induced transcriptional responses when growth is inhibited under standardized conditions.

### 3.7. Two Genes Upregulated by All Four Drug Treatments in U87MG

We next examined more closely the few genes that were shared between treatments.

Only two genes in the Top 200 upregulated set were upregulated by all four U87MG treatments: *PNLIPRP3* and *FAM49A*.

The *PNLIPRP3* gene encodes the protein Pancreatic Lipase Related Protein 3 (LIPR3), a rarely studied gene as judged by PubMed citation, previously observed as overexpressed in hepatocellular carcinoma [34]. We note from the literature that the peptide glioma growth inhibitor hHSS1/C19orf63/EMC10 also upregulates *PNLIPRP3* very highly in U87 cells [35]. From its entry in the Human Protein Atlas [36], *PNLIPRP3* has not been associated with either a favorable or unfavorable prognosis in glioma and is not expressed even at low levels in most normal human tissues.

The encoded LIPR3 protein bears a signal sequence and is most likely secreted from the cell, suggesting it might possibly represent an informative circulating biomarker for GBM. LIPR3 also possesses the catalytic triad characteristic of the esterase active site, and shares 47% overall homology to human pancreatic lipase (LIPP). LIPP is an important drug target upon which much medicinal chemistry has been undertaken, culminating in the development of the lipase inhibitor Xenical (otherwise known as Orlistat) approved for obesity management, reviewed in [37].

The *FAM49A* gene, also known as *CYRI-A*, encodes CYFIP-related Rac1 interactor A, a highly conserved regulator of the small GTPase RAC1, to which it binds [38]. *FAM49A* is expressed in the brain where the protein regulates chemotaxis, cell migration and epithelial polarization [39]. In contrast to LIPR3/*PNLIPRP3*, CYRIA*/FAM49A* is widely expressed in both normal and cancerous tissues and is a marker for unfavorable prognosis in both renal and urothelial cancer [40]. The X-ray structure of the closely related CYRIB protein has recently been solved [41], opening the way to homology modeling and structure-based drug design for CYRIA, if required.

### 3.8. Which U87MG Genes Are Upregulated in Mardepodect and Regorafenib Treatments?

As mentioned previously, we were especially interested in comparing the drug responses of GBM cells to the two clinical compounds Mardepodect and Regorafenib.

Including *PNLIPRP3* and *FAM49A*, U87MG cells treated with these compounds share 34 of their top set of 200 upregulated genes (Table 1). This set of 34 genes is particularly striking, encoding several cell membrane-associated proteins (GPNMB, CLC2D/*CLEC2D*, GCNT3, T4S19/*TM4SF19*, UNC5B, RFTN2, PXDC2/*PLXDC2*), cytoplasmic metabolism-related proteins (F262/*PFKFB2*, IDHC/*IDH1*, CGL/*CTH*), and transcription regulators (CRYM, MUSC/*MSC*, NUPR1, FWCH1/*FLYWCH1*, PRGC1/*PPARGC1A*), as well as secreted growth and cell guidance factors (GDF15, SLIT3, KCP, TSP2/*THBS2*). Several kinases associated with cancer cell survival, including KS6A2/*RPS6KA2* and PLK2, are also amongst this gene set.

We also see upregulation of the gene encoding growth/differentiation factor GDF15 in both Mardepodect and Regorafenib treated U87MG cells. GDF15 is of potential diagnostic and therapeutic significance in GBM, since elevated levels of GDF15 in the cerebrospinal fluid are associated with worse GBM outcome [42,43]. Conversely, downregulation of GDF15 increases T-cell infiltration into GBM tumors, improves immune responses and prolongs survival [44]. Reducing GDF-15 production and signaling have been proposed as ways of improving outcomes more generally in immunotherapy [45].

Also of potential clinical importance is the upregulation of the gene encoding the pseudokinase Tribbles homolog 3 (TRIB3). Like GDF-15, TRIB3 upregulation is associated with poor prognosis in GBM [46]. In ovarian cancer, TRIB3 downregulation inhibits progression via the MEK/ERK signaling pathway [47]. TRIB3 has also been reported to facilitate GBM progression, both by suppressing autophagy [46], and enhancing stemness [48].

### 3.9. Pathway Enrichment Analysis for the Upregulated U87MG Gene Sets

To gain further insight into the possible functional significance of the DIGEX data, pathway enrichment analyses were performed on the Top 200 upregulated gene sets accompanying individual drug treatments, using the Reactome database (Appendix A). Although mainly based on studies in non-GBM cell systems, such pathway enrichment analyses can highlight important gene networks that are shared by GBM cells.

Several distinctive genes characterized the Mardepodect-upregulated pathway signature “PIP3 activates AKT signaling” in U87MG cells, including those encoding the EMT-promoting transcription factors SNAI1 (Snail) and SNAI2 (Slug), which have key roles in tumor growth, invasion, and metastasis in GBM [49,50]. Again, highlighted within this signature is the pseudokinase *TRIB3* (Tribbles homolog 3), discussed previously in the context of the upregulated genes shared by Mardepodect and Regorafenib. All three members of the NR4A nuclear receptor gene family (*NR4A1*, *NR4A2*, and *NR4A3*) are also present within the Mardepodect “generic transcription” pathway signature, together with two members of the Ras related GTP binding (RRAG) gene family (*RRAGC* and *RRAGD*).

In contrast, Regorafenib-treated U87MG cells show prominent pathways for the “Response of EIF2AK1 (HRI) to heme deficiency”, “Netrin-1 signaling”, “Serine biosynthesis”, and “Transcriptional activation of mitochondrial biogenesis” pathways (Appendix A). The “Response of EIF2AK1 (HRI) to heme deficiency” pathway signature contains the component genes *DDIT3*, *TRIB3*, *ATF3*, and *ASNS*, a grouping associated with endoplasmic reticulum (ER) stress, and characteristic of genotoxic agents [51]. DDIT3 is a member of the CCAAT/enhancer-binding protein (C/EBP) family of transcription factors. It also features prominently amongst the most highly upregulated pathways in Fucoxanthin treated U87MG cells. Importantly, in glioblastoma the ATF4-ATF3-DDIT3 axis also triggers G2/M arrest [52].

Targeting energy metabolism has been suggested as a fruitful therapeutic strategy in GBM [53]. *PPARGC1A*, the gene encoding PRGC1, a transcriptional coactivator regulating energy metabolism via multiple transcription factor interactions, including the cAMP response element binding (CREB) protein and nuclear respiratory factors (NRFs), is a component of the “mitochondrial biogenesis” pathway. This pathway also contains TBL1X, an F-box-like protein involved in the recruitment of the ubiquitin/19S proteasome complex to nuclear receptor-regulated transcription units [54].

The “netrin signaling” pathway includes Netrin-4 (NET4/*NTN4)*, a specific netrin family member previously reported to promote GBM proliferation through ITB4/*ITGB4* signaling [55]. Netrins are laminin-related proteins that function in axon guidance and neurite growth and migration, tumorigenesis, angiogenesis and neural cell adhesion to endothelial cells, processes that are known to occur in GBM [56].

Upregulated alongside Netrin-4 is *UNC5B*, the gene which encodes the netrin receptor. In the absence of netrin, UNC5B triggers apoptosis, but an excess of netrin promotes cell survival, inducing interaction of UNC5B with the brain specific GTPase PIKE-L which opposes apoptosis by activating nuclear PI3K [56]. This interaction triggers activation of PI3K-signaling, prevents UNC5B’s pro-apoptotic activity and enhances neuronal survival. Studies of cell survival in glioma show that netrin acts as a pro-survival ligand for UNC5B in glioma as well [57], while also promoting invasion and angiogenesis of GBM cells by activating RhoA, cathepsin B, cAMP response element binding protein, and Notch signaling [58,59]. The genes encoding the secreted proteins ABLM1, ABLM3, and SLIT3, implicated in cell guidance and migration, are found within the same netrin cluster.

Taken together, the upregulation of these pathways by Regorafenib treatment indicates a delicately balanced network of cell proliferation and invasion.

Inspection of the pathways upregulated by LY-294002 and Fucoxanthin show the emergence of several new themes, including ‘signaling by interleukins’. In LY-294002-treated U87MG cells, we see upregulation of the interleukin pathway genes *RPS6KA5*, *IL36B*, *DUSP4*, *GAB2*, *FOS*, *PELI2*, *PTGS2*, *STX3*, *CCL20*, *IRS2*, *MAP3K8*, *SQSTM1*, *FOXO1*, *MMP1*, and *SOD2*, while the secretory chemokines *CCL20*, *CXCL8*, *CXCL1*, and *CXCL2*, component genes of the “Interleukin-10 signaling” pathway, show enhanced upregulation in Fucoxanthin-treated U87MG cells.

### 3.10. Downregulated Genes Revealed by DIGEX

Amongst the genes in U87MG downregulated by the four growth inhibitory compounds, 36 are shared within the top set of 200 most highly downregulated genes in each treatment (Figure 3 and Table 2). This is in sharp contrast to the upregulated genes, where only two genes were upregulated by all four treatments (Figure 2). The majority of the U87MG downregulated genes shared between treatments are associated with cell division, suggesting a coordinated and specific downregulation of the transcription of cell division genes in response to growth inhibition by these four compounds.

Amongst the most downregulated U87MG genes in all treatments is that encoding the transcription factor E2F8, the master regulator of the cell cycle [60]. Downregulation of E2F8 has been reported as a driver for prostate cancer growth suppression [61], and if cancer selectivity could be obtained, might represent a good target for stabilizing growth inhibition in GBM.

Other prominently downregulated genes which might encode good drug targets include RIR2/*RRM2*, which encodes the regulatory subunit M2 of ribonucleotide reductase, the enzyme that catalyzes the biosynthesis of deoxyribonucleotides for DNA synthesis. RIR2 is specifically inhibited by hydroxyurea and has been suggested as a combination therapy with temozolomide for GBM [62].

Further substantially downregulated genes include those encoding the transcription factor TCF19, which is associated with cancer cell survival and proliferation [63], and *FAM111B*, which encodes the DNA replication-associated serine protease F111B associated with both proliferation and cell cycle control [64,65]. Many genes encoding histones, important in maintaining nuclear and chromosome structure during cell cycling and division, are also significantly downregulated.

The mechanism controlling such a marked downregulation of genes encoding nuclear structural proteins after drug treatment is unclear. It is possible that mRNAs encoding nuclear components are no longer required in non-proliferating GBM cells and simply decay. Alternatively, the cells may be undergoing a controlled program of transcriptional and translational rebalancing, in which survival processes predominate and translation of cell division genes is specifically downregulated. Such dysregulation and restoration of translational homeostasis has been reported in fragile X syndrome where mRNA stability is thought to play a central role [66]. In either case, a new transcriptional equilibrium is being established, influenced by the presence of Mardepodect within the cells.

### 3.11. Pathway Analysis for the U87MG Downregulated Gene Sets

Pathway analyses were also performed for the U87MG downregulated gene sets, shown in Appendix A. These confirmed that all four compounds exerted anti-mitotic effects in U87MG cells, but also highlighted specific genes within these pathways. For example, members of the *MCM* (mini-chromosome maintenance) gene family are broadly downregulated within cells treated with all four compounds, while pathways downregulated by Fucoxanthin often include *SKP2* as a prominent component. SKP2 is a member of the F-box family of SCF ubiquitin ligases, pointing to reprogramming of the ubiquitin system during Fucoxanthin treatment.

### 3.12. Summary of the U87MG Results

Taken together, these DIGEX and pathway analyses highlight the considerable transcriptional plasticity of U87MG cells, with the upregulation of specific tumor cell survival pathways accompanying the downregulation of genes controlling mitosis and cell division. This pattern is consistent with transcriptional reprograming leading to a quiescent and/or drug resistant tumor cell population, a previously recognized mode of targeted therapy evasion [14,67].

### 3.13. Which DIGEX Genes Are Shared between Mardepodect Treated U87MG, T98G, and A172 Cells?

The U87MG DIGEX profiles for Mardepodect were then compared to those identified in T98G and A172 cells, to search for common targets (and identify specific differences) between the three cell lines. An overview of the Top 200 most highly induced genes within each cell type is shown in Figure 4, with individual genes listed in Table 3.

#### 3.13.1. Upregulated Genes

Our first observation was that over 75% of the Mardepodect upregulated genes in the three cell lines were cell-specific, suggesting that each GBM cell line responds in a unique way to Mardepodect treatment. A complete list of cell-specific genes is provided in Appendix A.

In contrast, very few upregulated genes were shared between the three Mardepodect-treated cell lines. Only three genes were expressed in common within the Top 200 upregulated gene sets: *GDF15*, *DUSP1*, and *SIK1*, all of which encode proteins that are involved in important growth-related processes.

GDF15 is a secreted growth factor, reportedly overexpressed in the cerebrospinal fluid (CSF) of GBM patients with poor treatment outcomes [42]. GDF15 binds to the GFRAL/RET receptor complex, stimulating cell growth through the ERK and AKT signaling pathways [68]. GDF15 has been suggested as a tumor-associated clinical biomarker suitable for liquid biopsy detection [69].DUS1/*DUSP1* is a dual specificity phosphatase which dephosphorylates and inactivates the MAP kinase MAPK1/ERK2, leading amongst other effects to aberrant regulation of the cell cycle. DUS1 plays important roles in the initiation, progression, and recurrence of GBM [70].SIK1 is a serine/threonine protein kinase that regulates transcription by phosphorylating transcriptional coactivators such as the CRTCs and HDACs. When cAMP levels increase, SIKs are phosphorylated by activated PKA and sequestrated by phosphorylated 14-3-3 proteins as inactive complexes in the cytoplasm [71]. Increased *SIK1* transcription in Mardepodect-treated GBM cells may reflect changes in these cAMP-driven processes.

When the three Mardepodect-treated cell types were analyzed pair-wise, more extensive correlations were revealed.

Strikingly, both A172 and T98G cells upregulated many genes associated with the cholesterol/isoprenoid biosynthesis pathway, including *HMGCR*, *IDI1*, *CYP51A1*, *FDFT1*, *MVD*, *HMGCS1*, *DHCR7*, and *INSIG1*. In parallel, but at a lower abundance, several genes involved in fatty acid metabolism were also upregulated, namely *DDIT4*, *FASN*, and *SCD*. These changes in lipid biosynthesis and metabolism are consistent with enhanced sterol and fatty acid biosynthesis, perhaps associated with autophagy [72].

Upregulated gene expression in T98G and U87MG cells showed many commonalities. Most noticeably, all members of the NR4 nuclear receptor transcription factor family—*NR4A1*, *NR4A2*, and *NR4A3*—were upregulated in both T98G and U87MG by Mardepodect treatment, as were the transcriptional repressor *HES1*, the epithelial to mesenchymal (EMT) transactivator *SNAI1*, and the histone methylation reader *ZCWPW2*. Several genes encoding members of the MAPK-signaling system were also upregulated in T98G and U87MG cells, including: the transcriptional and immune response regulator *TCIM (C8orf4)* which positively regulates G1-to-S-phase transition in the cell cycle, and promotes cell proliferation and inhibits apoptosis in thyroid and lung cancer [73,74]; the serine/threonine-protein kinase *SGK1* which also regulates cell growth, proliferation, survival, migration, and apoptosis through phosphorylation of MAPK1/ERK2, and interaction with MAP2K1/MEK1 and MAP2K2/MEK2; and the adapter protein TRIB1 which regulates COP1 ubiquitin ligase and MAP kinase signaling.

Genes controlling other processes such as complement decay (*CD55*), as well as cell adhesion, migration and hyaluronan degradation (*ITGB3*, *CEMIP*, *TNFAIP6*), were also upregulated.

Levels of the genes encoding 2 well-characterized druggable targets, KS6A2/*RPS6KA2* and the G-protein coupled receptor (GPCR) S1PR1, were also elevated.

KS6A2/*RPS6KA2*, also known as RSK/RSK3, is a member of the RSK serine/threonine-protein kinase family that acts as a downstream effector of ERK in the MAPK1/ERK2 and MAPK3/ERK1 signaling pathway, mediating cellular proliferation and survival in prostate cancer [75]. The related RSK kinase, KS6A3/RSK2, encoded by *RPS6KA3*, has been reported to regulate growth and invasion in GBM [76].S1PR1 is the GPCR for the bioactive lyso-sphingolipid sphingosine 1-phosphate (S1P) which is coupled to the G_i_ subclass of heteromeric G proteins. In cancer cells, signaling through S1PR1 leads to the activation of RAC1, SRC, PTK2/FAK1, as well as MAP kinases, and influences cell proliferation and survival in GBM [77].

Three further upregulated genes encode functionally relevant proteins: the small GTPase RND3; the mediator of E2F1-induced apoptosis, GRAM4/*GRAMD4*; and the interleukin IL6, which participates in an important axis for intrinsic VEGF production [78].

Comparison of the Mardepodect-upregulated genes shared in U87MG and A172 reveals further new signatures, with genes encoding the key glioma-associated cell surface proteins GPNMB and T4S19 (*TM4SF19*) being upregulated in concert with the anti-apoptotic heme-degrading enzyme HMOX1, reported to facilitate glioma survival and progression [79].

Interestingly, while the *FAM49A* gene is induced in the Top 200 genes expressed in Mardepodect-treated U87MG and A172 cells, the other gene induced by all four compounds in U87MG cells, namely *PNLIPRP3*, is notably absent. *PNLIPRP3* appears to be specifically induced in drug-treated U87MG cells.

#### 3.13.2. Downregulated Genes Shared between Cells

Cell-specific gene signatures are also seen within the Mardepodect downregulated genes (Appendix A), with a small number of highly downregulated genes occurring in all three cell lines, namely those encoding the nuclear serine protease F111B/*FAM111B*, the chemokine CCR2 receptor ligand CCL2 and the secreted carboxypeptidase CBPA4/*CPA4*. As proteases, both F111B and CBPA4 are druggable targets. CBPA4 is secreted as a zymogen, raising the further possibility of multi-level targeting during its maturation. The CCL2/CCR2 signaling axis is particularly relevant as a therapeutic target since its downregulation inhibits glioma development [80,81].

### 3.14. Differential Gene Expression Is Recapitulated in the Corresponding Pathway Analyses

Pathway analyses using the Reactome database were then undertaken to further examine the changes in individual gene expression profiles observed between the cell lines. They confirmed the striking divergence in signaling between U87MG and the other two cell lines.

In Mardepodect-treated U87MG cells, PIP3/AKT- and PTEN-driven signaling pathways were highly upregulated (Appendix A), reflected in upregulation of genes encoding the transcription regulators RRAGC/RRAGD and SNAI1/SNAI2 associated with these pathways. Enhanced PI3K signaling has previously been reported in this cell line [82].

In contrast, in both T98G and A172 cells, Mardepodect treatment prominently upregulated sterol biosynthesis pathways. Although these pathways are driven by the Sterol Response Element-Binding Proteins (SREBPs/SREBFs) [83], levels of the genes for the transcription factors SRBP1 (*SREBF1*) and SRBP2 (*SREBF2*) themselves were not elevated upon Mardepodect treatment, consistent with the post-translational regulation of these proteins by protein processing [84].

Differences in pathway upregulation in the U87MG compared to the T98G and A172 cell lines most likely reflects both their origins and stage of differentiation, as well as the more generally heightened transcriptional plasticity of U87MG cells. These pathway profiles may be useful diagnostics for analyzing GBM drug-response phenotypes in the clinic.

Cell-specific differences are also seen amongst the pathways downregulated by Mardepodect (Appendix A). The prominent cell cycle and cell division pathways characteristically downregulated in U87MG cells, are replaced in both T98G and A172 cells by immune-type cytokine signaling. Further downregulation of these pathways (and genes expressed within them) could be fruitful therapeutic targets.

To visualize functional connectivities within cell-specific DIGEX signatures, network analysis was undertaken for each signature. Again, highly significant differences between the cell lines were seen (Appendix A).

Taken together, these results demonstrate that a single drug treatment (in this case Mardepodect) elicits quite different gene expression responses in specific GBM cell lines. If these DIGEX signatures translate to freshly isolated patient-derived GBM cells, both the signatures themselves as well as the component drug targets within them, could form the basis for new personalized GBM treatment strategies.

### 3.15. How Do the DIGEX Signatures of Mardepodect-Treated Cells Compare to Those Seen in Regorafenib-Treated Cells?

An analysis of the Top 200 genes upregulated by Mardepodect in the three cell lines has been shown in Figure 4 and Table 3—only *GDF15*, *DUSP1*, and *SIK1* were coordinately upregulated in all three cell lines. Likewise, only three downregulated genes were shared in Mardepodect treated cells—C*PA4*, *FAM111B*, and *CCL2*. The small number of shared genes suggests that the drug response elicited by Mardepodect in the three cell lines is pleiotropic, involving the expression and recruitment of a wide variety of downstream signaling effectors, an observation confirmed by pathway analysis (Appendix A).

In sharp contrast, there was extensive overlap in both upregulated and downregulated gene expression profiles in the three cell lines treated with Regorafenib, with 30 upregulated genes and 41 downregulated genes shared within the Top 200 expressed genes (Figure 5 and Table 4).

Only *GDF15* was shared within the upregulated gene sets in Mardepodect and Regorafenib treated cells; the genes *DUSP1* and *SIK1*, seen previously in cells treated with Mardepodect alone, were absent, even within pairwise cell line comparisons.

Similar disparities were noted amongst the gene sets downregulated by Mardepodect and Regorafenib. Three genes—*CPA4*, *FAM111B*, and *CCL2*—were observed as downregulated in all three Mardepodect-treated cell lines (Figure 4), but only one of these, *FAM111B*, was seen in the Top 200 downregulated genes in Regorafenib-treated cells. *CPA4* was entirely absent from any of the Top 200 gene sets downregulated by Regorafenib, and *CCL2* was only downregulated in the A172/T98G pairwise comparison.

To summarize, the three GBM cell types differ markedly in their DIGEX profiles when growth is inhibited by Mardepodect and Regorafenib under standardized conditions, yielding highly informative and distinctive drug response signatures.

### 3.16. What Can Be Inferred from the Cell-Specific DIGEX Signatures?

Finally, we examined the cell line-specific gene sets from each treatment. Results obtained by treating U87MG cells with all four inhibitors showed many compound-specific genes (Figure 2 and Table 1), with the two clinical candidates Mardepodect and Regorafenib each showing over 100 compound-specific genes amongst the Top 200 examined.

Extending these observations to all three GBM cell lines, we again see expression of a high level of cell-specific genes (Appendix A for Mardepodect; Appendix A for Regorafenib). Network analysis confirmed the marked differences in DIGEX profiles between treated cell lines (Appendix A).

In summary, taking the Top 200 genes in each DIGEX profile operationally defines a distinctive drug ‘fingerprint’, specific to each cell line, from which much underlying biological information can be retrieved.

### 3.17. Which Genes Encode Proteins That Could Be Viable Drug Targets in GBM?

The development of new GBM therapies requires target validation to be coupled to effective drug and vaccine production. In this study, we have used genome-wide gene expression analysis of drug-treated cells to reveal subsets of genes which are characteristic of the underlying cell biology of the GBM cells. These DIGEX signatures are powerful diagnostics—but how many of the genes thus identified represent viable drug targets?

Complete profiles of the Top 200 differentially expressed genes drawn from the set of 18,316 tracked in all three GBM cell lines studied, are shown for all treatments in Appendix A. These gene sets contain many biological targets that have not previously been directly associated with GBM, including specific adhesion molecules, transcription factors, protein kinases, and glycosyltransferases. We used the Pharos, NCBI Gene, and UniProt databases to identify, classify, and annotate all the likely drug and vaccine targets present in the three cell lines. In the analysis below, UniProt protein entries are denoted in block capitals with corresponding NCBI Gene entries in italics.

The Pharos database is a chemical biology resource that allows rapid association of genes encoding proteins with potential chemical modulators, including both approved drugs and exploratory compounds. We use the database as part of a drug target triage strategy, separating the Top 200 gene sets into groups encoding (1) proteins with associated FDA approved drug modulators; (2) proteins with chemical modulators that can be used as starting points for drug discovery; and (3) proteins which merit further biological study.

#### 3.17.1. U87MG Cells Treated with Mardepodect

To illustrate this process, focusing on the cell line U87MG treated with Mardepodect, we see that one of the most highly upregulated genes is *PNLIPRP3*, encoding the lipase LIPR3, previously seen as one of only two genes that are shared within the Top 200 most highly upregulated genes in all four initial drug treatments in this cell line (Figure 2).

The *PNLIPRP3* gene encodes a novel druggable protein, with a well-defined Lipase domain harboring the active site, sandwiched between an N-terminal signal peptide and a hydrophobic PLAT domain—a structure the gene shares with pancreatic lipase (LIPP/*PNLIP*) and the other members of this gene family (LIPR1/*PNLIPRP1* and LIPR2/*PNLIPRP2*). However, LIPR3 has no specific FDA-approved or exploratory chemical leads associated with it, and is therefore annotated by Pharos as Tdark, implying a protein without well-defined biological precedent as a drug target and without chemical leads [85].

Although no drug discovery campaigns have been reported for LIPR3, the pancreatic lipase gene family of which LIPR3 is a member has been the subject of considerable medicinal chemistry attention due to the role of the closely related LIPP protein in obesity, for which there is an FDA-approved drug (Orlistat) with an associated X-ray co-crystal structure [37].

*PNLIPRP3* is highly induced only in drug treated U87MG cells and is not highly upregulated by Mardepodect in either T98G or A172 cells (Appendix A). It is not even modestly expressed in normal tissues [36]. We classify LIPR3/*PNLIPRP3* as a novel druggable GBM target in cells with the U87MG phenotype.

Using Pharos classification alone as a benchmark, amongst the Top 200 U87MG genes upregulated by Mardepodect, we identify 36 (18%) as encoding potential targets with currently unexplored biology; 118 (59%) as target genes corresponding to proteins for which biological targeting rationales exist but which have no associated chemical modulators; 36 genes (18%) encoding proteins with exploratory chemical leads; and only 7 genes encoding proteins with corresponding FDA-approved drugs.

The FDA-approved target class is important, since drugs targeting these proteins could be repositioned immediately within clinical trials in GBM. The relative paucity of validated FDA-approved drugs for the targets we reveal by DIGEX suggests that target validation remains a key challenge for GBM drug discovery.

For Mardepodect-treated U87MG cells, the 7 upregulated FDA-approved targets as classified by Pharos comprise: the Vitamin D-binding nuclear receptor VDR; the secreted cytokine IL6; the Thioredoxin Reductase TRXR1/*TXNRD1*; the kinase-insert domain receptor VGFR2 (known variously as *KDR*, *FLK1*, *CD309*, *VEGFR*, or *VEGFR2*), the integrin beta chain ITB3/*ITGB3*, and the GPCRs S1PR1 and EDNRA.

#### 3.17.2. T98G and A172 Cells Treated with Mardepodect

Broadening this analysis from U87MG to the other two GBM cells T98G and A172, we see that very few of the U87MG Pharos-annotated FDA-approved drug target genes are replicated within the Top 200 genes upregulated by Mardepodect, translating into a gene signature with a radically different FDA-approved drug profile.

For T98G cells, 12 genes form the ‘FDA signature’, comprising the two interleukins *IL1B* and *IL6*; five G-protein coupled receptors (the dopamine receptor *DRD2*, the adrenoceptor *ADRB2*, the adenosine receptor *ADORA1*, the sphingosine-1-phosphate receptor *S1PR1*, and the bradykinin receptor *BDKRB2*); the heparin-binding growth factor *VEGFA*; two cholesterol biosynthesis enzymes *HMGCR* and *HSD11B1*; the phosphodiesterase *PDE4D*; and, as in U87MG cells, the integrin beta chain *ITGB3*. Except for *ITGB3* upregulation, the U87MG and T98G FDA drug signatures are mutually exclusive.

For A172 cells, the FDA signature includes nine components: the two cholesterol biosynthesis enzymes *HMGCR* and *FDPS*; the phosphodiesterase *PDE7B;* the somatostatin receptor *SSTR2;* the androgen receptor *AR* (also known as the nuclear receptor *NR3C4*); the NMDA receptor *GRIN2A*; the carbonic anhydrase isozyme *CA12;* the thyroid hormone receptor *THRA*; and the lipase *LIPF*. Again, the U87MG and A172 FDA drug signatures are mutually exclusive. T98G and A172 cells notably share *HMGCR* expression.

Turning our attention from FDA targets to the more extensive Exploratory drug target class exemplified within Pharos, we find many attractive, chemistry-led drug discovery targets.

For U87MG, these include the GPCRs T2R14/*TAS2R14*, C3AR/*C3AR1*, C5AR1, GPR84, and GP183/*GPR183*; the kinases SIK1, PDK4, ACKR3, PLK2, KS6A2/*RPS6KA2*, F262/*PFKFB2*, SGK1, KITM/*TK2*, and CHKA; the interleukin IL1A; the metalloproteinases ATS5/*ADAMTS5* and MMP14; the nuclear receptor NR4A2; the asparagine N-linked glycosyltransferase TUSC3; the cell adhesion molecule KIAA1462/JCAD; the transporters ACATN/*SLC33A1*, XCT/*SLC7A11*, NRAM2/*SLC11A2*, and CLCN7; the lipases LIPE/*LIPG* and LIPR/*PNLIPRP3*; the enzymes CGL/*CTH*, DHB14/*HSD17B14*, HMOX1, OGT1/*OGT*, AK1C1/*AKR1C1*, FHIT, and IDHC/*IDH1*; the dual specificity phosphatase DUS1/*DUSP1*; the ligand-gated chloride channel GBRR1/*GABRR1* (otherwise known as the GABA(C) receptor); the G-protein linked potassium channel KCNJ3; and the phosphocholine/phosphoethanolamine phosphatase PHOP1/*PHOSPHO1*.

Several previously validated anti-proliferative targets are contained within this U87MG Exploratory target set, including:The atypical chemokine GPCR ACKR3/CXCR7, which in glioma cells transduces signals via the MEK/ERK pathway, mediating resistance to apoptosis and promoting cell growth and survival [86]; andThe nuclear receptor NR4A2, previously validated as a drug target in glioblastoma [87].

Mardepodect-inhibited A172 and U87MG cells share some of these Exploratory targets, including LIPE/*LIPG*, HMOX1, GP183/*GPR183*, DUS1/*DUSP1*, NRAM2/*SLC11A2*, and the serine/threonine protein kinase SIK1. Druggable components of the cholesterol/fatty acid biosynthesis pathways are also prominent within the A172 Exploratory target set (DHCR7; HMCS1/*HMGCS1;* FDFT/*FDFT1*; KIME/*MVK;* MVD1/*MVD*; IDI1; DHB7/*HSD17B7*; CP51A/*CYP51A1*; ACACA; SCD; ELOV6/*ELOVL6;* FAS/*FASN*; ABHD6; FABPH/*FABP3*), together with more established anticancer drug discovery targets such as the PI3-kinase P3C2B/*PIK3C2B* and the lysine-specific demethylase KDM4D. We also see unique targets, such as the small GTPase RAB7L/*RAB29* (involved with LRRK2 in vesicle trafficking); the P2X receptor P2RX7; the cell adhesion protein VITRN/*VIT*; the opioid neuropeptide GPCR OPRX/*OPRL1* and the olfactory GPCR OR1L4; the spermatogenesis-associated, calmodulin-binding protein SPT17/*SPATA17*; the ectonucleoside diphosphatase ENTP1/*ENTPD1*; the cystathionine beta-synthase CBS; the DNA methylation enzyme DNM3B/*DNMT3B*; the P-type ATPase AT12A/*ATP12A*; the putative P-glycoprotein-associated drug transporter EBP; the apoptosis suppressor XIAP; the protein tyrosine phosphatase PTN22/*PTPN22*; and the ephrin receptor tyrosine kinase EPHA8.

In passing, for target-based drug discovery purposes, it is notable that several specific members of extended gene families of potential drug discovery importance are revealed by DIGEX in these exploratory gene sets, for example, the PI3-kinase catalytic domain isoform P3C2B/*PIK3C2B* and the lysine demethylase KDM4D. PI3-kinases participate in the signaling pathways involved in cell proliferation and oncogenic cell survival, and the induction of P3C2B/*PIK3C2B* is therefore not surprising, since this protein has previously been identified as significantly correlated with cellular resistance to erlotinib [88]. However, Regorafenib also upregulates P3C2G/*PIK3C2G*, confirming this isoform, also, as a potential target in drug resistant GBM [89]. Likewise, although KDM5A has previously been identified in temozolomide resistant GBM cell lines [90], here we see Mardepodect upregulating KDM4D and KDM7A. The advent of small molecules specifically targeting individual lysine demethylase isoforms may open the way to more precise drug targeting within this extended family [91,92].

Many of the highly upregulated Exploratory targets seen in A172 cells are also present in T98G, including GTR3/*SLC2A3*; DUS1/*DUSP1*; the sterol biosynthesis enzymes HMCS1/*HMGCS1*; SCD; FDFT/*FDFT1*; DHCR7; MVD1/*MVD;* IDI1 and the multi-functional fatty acid biosynthesis enzyme FAS/*FASN*, as well as the additional steroid hormone biosynthesis enzymes 3BHS1/*HSD3B1* and DHB2/*HSD17B2* and ERG1/*SQLE*, the rate-determining enzyme in the steroidogenic pathway. Several highly upregulated Exploratory targets seen in A172 cells are also shared with U87MG: NR4A2; SGK1; KS6A2/*RPS6KA2*.

Top 200 Mardepodect-upregulated Exploratory targets seen only in T98G include the IGF-binding protein IBP4/*IGFBP4;* the membrane lipid remodeling phospholipase PA24A/*PLA2G4A;* the chromatin silencing histone H10/*H1F0;* the prostaglandin transporter SO2A1/*SLCO2A1;* the drug metabolizing methyltransferase NNMT; the histone demethylase KDM7A; the neurotrophin receptor signaling adapter BEX1; the transcription factor JUN; the hypoxia-inducible master transcription activator HIF1A; the serine/threonine protein kinases TNI3K/*TNNI3K* and NIM1/*NIM1K*; the MMP-9 activator MMP26; the histamine receptor GPCR HRH4; the tyrosine protein kinase FRK; and the GPCR specific serine/threonine kinase GRK5.

Together, these targets represent a prodigious amount of relatively unexplored drug discovery space. Systematic target validation studies are now required to establish experimentally how many of the exploratory targets within the Mardepodect-induced GBM cell phenotypes are valid as drug targets for GBM. We illustrate features of a possible chemical biology-driven target validation process below.

### 3.18. Downregulated Genes May Indicate Cell Cycle Control Imposed by Drug Treatment

The Pharos analysis of the genes that are coordinately downregulated by Mardepodect in the three GBM cell lines, is presented in Appendix A. As observed in the case of U87MG, above, these DIGEX profiles are dominated by reduced cell cycle and cell division gene expression.

Although a full analysis of the downregulated genes accompanying drug treatment is beyond the scope of this initial DIGEX study, it is clear from initial inspection that the downregulated gene sets contain many intriguing drug discovery targets, from well characterized enzymes such as ribonucleoside-diphosphate reductase RIR2/*RRM2*, to less well-known targets such as the serine protease F111B/*FAM111B*. The Venn analyses also show that most of the downregulated cell-cycle associated genes are differentially regulated between GBM cell types and the four drug treatments, suggesting a tight and precise downregulation of cell division, rather than random repression, possibly reflecting a more complex spatiotemporal control of cell division [93].

### 3.19. Summary of Drug-Induced Gene Expression (DIGEX) Analysis Results

An overall diagrammatic summary of the most highly upregulated drug-induced genes encoding potential drug targets, secretory proteins and cell surface antigens expressed in the three GBM cell lines is shown in Figure 6.

### 3.20. Validating Individual Drug-Inducible Genes as Pharmacological Targets in GBM as Monotherapies and Drug Combinations

DIGEX profiling in conjunction with Pharos yields a rich vein of potential targets. With detailed bioinformatic analyses in hand, we moved to experimental validation of some of the targets themselves as potential GBM modulators.

To establish a screening sequence for potential combination therapies using Mardepodect as the initial drug, we identified a set of targets with cognate inhibitors to validate our triage strategy. These included:Two FDA approved inhibitors for HMGCoA reductase, Atorvastatin and Simvastatin. The gene encoding HMGCoA reductase (*HMGCR*) is in the Top 200 genes upregulated by Mardepodect in A172 and T98G cells but not U87MG. *HMGCR* is absent from the Regorafenib Top 200.Two exploratory inhibitors of the salt-inducible kinase SIK1, HG-9-91-01, and WH-4-023. The gene encoding SIK1 is present in the Top 200 genes upregulated by Mardepodect in all three GBM cells but absent from the Top 200 in Regorafenib-treated cells.Two inhibitors of the Janus Kinase JAK2, the FDA approved drug Ruxolitinib and the exploratory compound AZD1480. The gene encoding JAK2 is induced in U87MG by LY-294002 [20] but absent from the Top 200 upregulated genes in both Mardepodect and Regorafenib treated cells.Two inhibitors of the bradykinin B2 receptor, Icatibant and WIN 64338. The *BDKRB2* gene is in the Top 200 upregulated genes in Mardepodect treated T98G cells.

### 3.21. Combinations of the PDE10A Inhibitor Mardepodect and Regorafenib

Combination therapy is a major objective in GBM [9]. Although the screening sequence deployed here (DIGEX-Pharos-Pharmacology) was designed to uncover new drug discovery targets, it might also predict target combinations for validation. We therefore also determined whether any of the compounds we had identified individually to inhibit GBM cell proliferation were effective in combination.

As an initial experiment, we tested the two clinical compounds Mardepodect and Regorafenib in combination. Synergy between PI3K and cAMP signaling pathways has previously been suggested as potentially relevant in GBM [94]. We observed synergy in U87MG cells (with a maximum synergy score of 24), but modest antagonism between Mardepodect and Regorafenib in both T98G and A172 cells (with a maximum score of −21) (Figure 7).

### 3.22. HMGCoA Reductase

Our data indicated that HMGCoA reductase (HMDH/*HMGCR*) was upregulated in the Mardepodect-treated A172 cell line where it was the 33rd most upregulated gene. HMDH/*HMGCR* has recently been suggested as a therapeutic target in GBM [83] and upregulation of HMDH/*HMGCR* has previously been shown to positively regulate the growth and migration of the GBM cell lines U251 and U373 [95].

Both Atorvastatin and Simvastatin markedly inhibited cell proliferation (Figure 8a,b), with both U87MG and T98G cells showing synergy with Mardepodect (Figure 8c–h). A172 cells showed higher sensitivity to statin inhibition but showed antagonism upon combination of the statins with Mardepodect. Both Atorvastatin and Simvastatin are safe, widely prescribed FDA approved drugs, belonging to the lipophilic statin class which carries no borderline risk of causing glioma [96].

Additional genes associated with cholesterol biosynthesis—including *INSIG1*, *HMGCS1*, and *RNF145*—are also upregulated by Mardepodect in at least one of the three GBM cell lines studied here, warranting further examination of the pathway and its rate-limiting steps as adjunct therapeutic targets in GBM.

### 3.23. Salt-Inducible Kinase SIK1

Salt-inducible kinase isoform 1 (*SIK1*) is highly upregulated in all three GBM cells treated with Mardepodect (Table 3). Acting through their interaction with 14-3-3 proteins, SIKs disrupt cAMP signaling, promoting inhibitory phosphorylation on CREB-regulated transcription coactivators [71,97].

We have previously shown that cAMP inhibits the growth of rat C6 glioma cells [20], and we know that the PDE10A inhibitor Mardepodect increases the levels of cAMP in the human GBM cell lines used here. Salt-inducible kinase (*SIK*) is also one of six genes associated with significantly shorter patient survival in GBM [98].

We used two commercially available compounds to investigate the effects of SIK inhibition on proliferation: HG-9-91-01, originally synthesized for studies of SIK involvement in inflammation [99], and WH-4-023, originally designed as a lymphocyte specific kinase (LCK) inhibitor but which has SIK inhibitory properties [100].

HG-9-91-01 is a pan-specific SIK inhibitor with an IC50 versus SIK1 of 0.92 nM, SIK2 of 6.6 nM, and SIK3 of 9.6 nM, with lower activity against the kinases NUAK2, SRC, LCK, YES, and BTK, and the FGFR and EphR families. WH-4-023 is also pan-specific towards the SIK family, with lower affinity on BTK, FGFR, JAK2, KDR, p38 alpha, SYK, TIE 2, and ZAP70.

Both inhibitors were anti-proliferative in all three GBM cell lines. The strong induction of *SIK1* upon Mardepodect treatment is consistent with GBM cells sensing elevated cAMP levels and raising *SIK1* mRNA levels to compensate for this. Considerably greater sensitivity to one of the SIK inhibitors, HG-9-91-01, was seen in A172 cells, but all three cell lines were inhibited by WH-4-023 to the same degree (Figure 9a,b).

Having seen marked inhibition of growth in the three cells by the two SIK inhibitors, we also tested Mardepodect in combination with the SIK inhibitor HG-9-91-01. Here, we observed strong synergy (with a synergy score rising to 40)—but only in T98G cells (Figure 9d). Both U87MG and A172 cells showed much lower synergy scores (Figure 9c,e).

The synergy between Mardepodect and HG-9-91-01 in T98G cells suggests an inducing drug (Mardepodect) synergizing with an inhibitor of an induced target (SIK1), with the two targets operating as a ‘node’ within the cAMP signaling pathway. Upregulation of induced targets within the cancer cell’s survival programs has been suggested to promote drug resistance [101].

*SIK1* is most upregulated by Mardepodect in U87MG cells, where it is the third most highly upregulated gene (Appendix A). One might imagine that U87MG cells would be more sensitive to SIK inhibition, but Mardepodect and HG-9-91-01 show only modest synergy in U87MG cells. In T98G cells, however, where strong synergy is seen, *SIK1* is only the 65th most upregulated gene, suggesting that in this case the degree of target upregulation alone does not predict combinatorial drug sensitivity.

A potentially important observation is that neither of the other two SIK subtypes, *SIK2* and *SIK3*, show pronounced induction upon Mardepodect treatment, implicating specific transcriptional selection of the *SIK1* subtype during drug response. Drug discovery focused on achieving SIK1 selectivity may therefore be a relevant objective in the design of effective GBM therapies. In this context, it is important to note that effective target validation by chemical biology is always dependent on the quality and selectivity of the chemical probes available [17,18].

As far as we know, no SIK inhibitors are in development as antiproliferative agents in GBM. However, topical SIK inhibitors are being developed as sunscreen agents in melanoma [102]. SIKs also mediate parathyroid hormone receptor activity in bone development and remodeling [103,104] and the inflammatory phenotype in activated myeloid cells [105,106]. SIK inhibitors, with their therapeutic potential for the treatment of inflammatory and autoimmune diseases, are thus a commercially attractive drug class which may possess the added advantage of being suitable for repurposing in GBM.

Our results confirm the general importance of the SIK pathway in GBM growth control. They also show that important growth mediators may be buried within DIGEX signatures, since *SIK1* is only the 41st most highly upregulated gene in Mardepodect-treated T98G cells and the 99th most highly upregulated gene in Mardepodect-treated A172 cells, despite being one of the most highly upregulated genes in Mardepodect-treated U87MG cells.

### 3.24. Janus Kinase JAK2

In previous studies, we observed upregulation of *JAK2* by LY-294002 in U87MG cells [20]. JAK2 plays a central role in phosphorylation of glioma-associated STAT3, a key component of the PI3K-signaling pathway [107]. The selective JAK2 inhibitors SAR317461 and AZD1480 have been reported to inhibit GBM proliferation via this pathway [108,109], and combining the approved EGFR inhibitors Erlotinib and Osimertinib with the JAK2 inhibitor AZD1480 induces irreversible apoptosis in GBM [110]. The JAK1/2 inhibitor, Ruxolitinib, is an approved drug for the treatment of polycythemia vera and myelofibrosis [111,112].

We examined the effects of both AZD1480 and Ruxolitinib on proliferation in the three GBM cell lines (Figure 10). Both compounds inhibit the growth of all three GBM cells. A172 cells showed greater sensitivity to AZD1480 than did U87MG, with T98G showing intermediate sensitivity, an order of potency previously observed by others for the JAK2 selective inhibitor SAR317461 [108].

We then tested the JAK2 inhibitor AZD1480 in combination with the prototypic PI3K inhibitor LY-294002 (Figure 10c–e). Here, strong synergy was seen in U87MG cells, with synergy scores rising to 54, with some synergy evident in T98G cells. No synergy was seen in A172 cells. Synergy was compound- and possibly subtype-specific, since the non-selective JAK1/2 inhibitor Ruxolitinib showed no synergy with LY-294002 in U87MG cells.

Subtype selectivity in targeting kinases such as JAK2 and SIK1 in GBM will most likely be important in the clinic since we know that JAK and SIK subtypes play important roles in processes such as macrophage differentiation, dendritic cell function and innate immunity [104,105,113]. Balancing subtype selectivity may thus provide benefits in GBM immune recognition as well as GBM growth inhibition.

### 3.25. GPCRs

We noted prominent and specific upregulation of several genes encoding GPCRs in the DIGEX data (Appendix A), including *ADORA1*, which encodes the adenosine A1 receptor (highly upregulated in T98G cells by Mardepodect); *BDKRB2*, which encodes the bradykinin B2 receptor (highly upregulated in T98G cells by Mardepodect); *DRD2*, which encodes the dopamine D2 receptor (upregulated by Mardepodect in T98G cells); and *GPR84*, which encodes the orphan GPCR GPR84 (the most highly Mardepodect-upregulated gene in U87MG cells).

#### 3.25.1. Bradykinin B2 Receptor (B_2_R)

The gene encoding the B_2_R, *BDKRB2*, was markedly upregulated in Mardepodect-treated T98G cells (Appendix A). B_2_R is an EMT-related biomarker and predicts poor survival in glioma [114], while bradykinin itself enhances invasion of malignant glioma into the brain parenchyma [115]. Pharmacological studies in the human astrocytoma cell line D384 have shown the B_2_R to be present and functionally linked to phospholipase C and inhibition of dopamine stimulated cyclic AMP accumulation [116].

Although the bradykinin receptor antagonist drug class represents an area of intense current drug discovery opportunity [117], only one bradykinin receptor ligand is currently used in clinical practice, the B_2_R antagonist Icatibant/HOE-140 [118].

We tested Icatibant in GBM cells alongside the nonpeptide bradykinin B2 antagonist WIN 64338 [119]. Consistent with reports that the synthetic peptide Icatibant is metabolically vulnerable, Icatibant proved inactive as a growth inhibitor in our cellular models of GBM proliferation (Figure 11a). In contrast, the non-peptide B_2_R antagonist WIN 64338 showed promising growth inhibitory activity against all three GBM cell lines (Figure 11b), confirming the potential involvement of B_2_R signaling in GBM proliferation.

Very interestingly, the T98G cell line, which showed marked upregulation of the *BDKRB2* gene after Mardepodect treatment, demonstrated pronounced synergy when Mardepodect was combined with WIN 64338 (Figure 11d). Neither U87MG or A172 cells showed comparable synergy (Figure 11c,e). Further examination of the comparative pharmacology of bradykinin signaling in these 3 GBM cell lines is merited.

#### 3.25.2. Other GPCRs

GPCRs are the most common class of drug target [120] and are key targets in oncology [121]. We observed several upregulated genes encoding GPCRs in both Mardepodect- and Regorafenib-treated cells, including G Protein-coupled Receptor 37 (*GPR37*) and the dopamine D2 receptor *DRD2*, both of which, like B_2_R, are Gi-coupled GPCRs [122,123]. GPR37 also acts as a cell survival factor [124], occurring as a complex with DRD2 at the cell surface where it exerts a neuroprotective effect [125,126].

Importantly, the DRD2 receptor is the target for the promising anticancer compound TIC10/ONC201 [127], currently in clinical trials for GBM [128]. Drug resistance to ONC201 has recently been observed [129], making pharmacological exploration of the signaling networks surrounding GPCRs such as GPR37, DRD2, AA1R/*ADORA1*, and BKRB2/*BDKRB2* of particular therapeutic relevance.

We have shown previously that increasing cAMP levels in glioma cells results in growth inhibition [29], suggesting that, in conjunction with the use of PDE inhibitors such as Mardepodect, raising intracellular cAMP levels in GBM cells by antagonizing Gi-linked GPCRs such as B_2_R could be therapeutically relevant.

Another GPCR which merits future pharmacological follow-up in GBM is the orphan GPCR, GPR84. In Mardepodect-treated U87MG cells, *GPR84* was the most highly upregulated gene (Appendix A). GPR84 is a fatty-acid binding protein involved in fibrosis [130] and is also essential in the maintenance of cancer stem cells in acute myeloid leukemia [131]. A range of ligands is available for exploring its function [132].

### 3.26. Summary of the Compound Synergy Studies

These initial results show that synergy is often cell-type specific, especially evident in the LY-294002/AZD1480 combinations which showed strong synergy in U87MG cells. Cell-type specificity was also seen in Mardepodect/HG-9-91-01 combinations, with T98G cells on this occasion showing strongest synergy. In general, A172 cells exhibited lower levels of synergy than either U87MG or T98G cells, although they often showed higher compound sensitivities. Combinatorial testing of further inhibitors guided by the DIGEX data may enable additional synergies to be found.

## 4. Discussion

Previous work from our laboratory has used genome-wide drug-induced gene expression (DIGEX) as a method of defining the molecular phenotypes induced by growth inhibition in the GBM cell line U87MG [20]. In that study, we focused on two compounds, the prototypic PI3K inhibitor LY-294002, a pan-PI3K inhibitor with well-characterized multiple molecular modes of action [133,134], and Fucoxanthin, a marine algal natural product with anti-cancer growth inhibitory properties [135]. We were somewhat surprised to see the great variation in gene expression produced by these two growth inhibitors, having imagined that a single cell line such as U87MG, upon growth inhibition, would display a more restricted drug-induced phenotype.

To investigate these observations further, here we have used further growth inhibitors, including two clinical antiproliferative drugs, the PDE10A inhibitor Mardepodect and the multi-kinase inhibitor Regorafenib, profiling them in two additional GBM cell systems. Our results reinforce our earlier observations of the pleiotropic nature of the DIGEX response, uncovering a multitude of new drug response pathways, with relatively few genes expressed in common between treatments with these two further drugs and the two earlier compounds.

We were also expecting to see extensive crossover in DIGEX phenotypes between different cells treated with the same drugs, which we would then be able to interrogate for drug discovery further through a target-directed process such as MIPS (Mechanism-Informed Phenotypic Screening) [16]. In fact, the DIGEX process unveils highly divergent cell-specific gene signatures, revealing a distressingly complex combinatorial landscape from which future drug discovery campaigns for this challenging cancer will need to be prioritized.

Amongst these signatures, we can see the specific induction of several druggable targets that have been implicated previously in GBM growth (e.g., SGK1, NR4A2, discussed above). However, many targets represent completely new avenues for GBM drug discovery (e.g., LIPR3/*PNLIPRP3* and SIK1).

Amongst these emerging drug targets, we have confirmed two protein kinases, JAK2 and SIK1, as antiproliferative targets in their own right. When inhibited, both show synergy, on a cell-specific basis, with the primary drug used to generate the original DIGEX target landscapes. It is our hope that targeting the emerging properties of such drug-treated systems will reveal further actionable drug combinations.

While genome-wide transcriptomics provides a powerful platform for evaluating drug response in GBM, genetic technologies such as CRISPR and transposon mutagenesis are providing complementary information on the role of cancer drivers in tumor evolution [136,137]. Combining information from both sources promises to redefine the target landscape for drug discovery in GBM.

The DIGEX results also raise the possibility that the extreme drug resistance observed in GBM is at least partially due to the wide range of survival pathways that growth inhibition induces in GBM cells, indicating both the need for a broader range of effective drugs than we currently possess, and the informed use of our current armamentarium within personalized treatment schedules.

Our analyses based on gene annotations within the Pharos database, show that the current spectrum of clinically approved drugs available to address the GBM target space we define here, is relatively limited. Moreover, many of these drugs have been optimized and approved for indications in which CNS penetration has been minimized. There is a pressing need for rigorous validation of new molecular mechanisms as effective drug targets in GBM before embarking on costly programs of lead optimization. Combination therapy guided by emerging GBM biology will shorten these odds.

At a pharmacological level, our results are intriguing. Mardepodect, as a PDE inhibitor, would be expected to influence target cells through cyclic nucleotide signaling. However, scrutiny of the upregulated gene sets induced by this compound does not reveal a sustained effect on the expression of central cAMP or cGMP regulating processes, such as those operating through adenyl- or guanyl-cyclases. We do see upregulation of the genes encoding specific PDEs (*PDE4B*, *PDE4D*, and *PDE7B)* in Mardepodect-treated cells, but not *PDE10A* itself. The reason for this may be that the transcriptomics approach is measuring the emergence of a new steady-state within the drug-treated tumor cell population, involving more subtle regulation of localized cAMP- and cGMP-driven phosphorylation networks, as reported for PDE10A-regulated systems in the spiny neuron [138].

The three GBM cell types employed in this study are widely used as established GBM cell-based model systems [33]. Using this limited cell palette, well-defined drug response profiles emerge, with each cell line exhibiting characteristic DIGEX signatures. The fact that these GBM cell lines retain the capability of forming infiltrative tumors upon xenografting, yet respond so differently to drug challenges, may indicate a multi-dimensional phenotype containing features that are stably frozen in developmental time, yet capable of resurrection given the appropriate environment or selection pressure.

Designing experiments to capture the dynamic behavior of such pleiotropic systems is difficult and requires further studies across the drug dose–response and treatment timeframes to pinpoint key effectors—the current dataset is only a preliminary step on the way to understanding and exploiting the complex coordination of the GBM cell’s drug response.

## 5. Conclusions

Glioblastoma (GBM) is a particularly challenging cancer, with few treatment options. Nevertheless, this dark landscape is gradually being illuminated by the application of powerful genomics technologies to define the molecular drivers underlying this complex cancer. Here, we have used transcriptomics, in the form of genome-wide drug-induced gene expression (DIGEX) analysis, to measure the changes in abundance of over 18,000 individual genes during drug treatment. We define these changes for three established GBM cell lines, identifying the key targets and pathways which these cells deploy to resist drug treatment. The good news is that we now have the genomic tools to interrogate and interpret these drug responses; the bad news is that the emerging landscape evoked in the cancer cell is both extremely complicated and highly pleiotropic.

Future therapy will rely heavily on diagnosing and targeting aggressive GBM cellular phenotypes, both before and after drug treatment, as part of personalized therapy programs. Using genome-wide information, we should be able to interrogate the therapeutic targets and signaling pathways emerging from this landscape, either individually, or at a phenotypic level, deploying integrated RNAi, CRISPR, chemical biology, and pharmacology platforms to identify and validate their relative importance to the proliferating GBM cell. Individual, personalized treatments will be essential if we are to address and overcome the pharmacological plasticity that GBM exhibits, and genome-wide DIGEX may represent a fast and comprehensive technology with which to validate and position future drugs and diagnostics for this challenging cancer.

One final thought. Although small molecule drug therapy has become central to the way we think of cancer treatment, alternative therapies—such as mRNA-directed vaccination and immunotherapy—are emerging as powerful treatment options. The GBM landscapes revealed by DIGEX and discussed here in the context of drug discovery, could well inform new diagnostics to support the development of multi-modal therapies, perhaps combining drug and vaccine treatment within personalized GBM therapy. A new age of phenotypic drug discovery?

## Figures and Tables

**Figure 1 cancers-13-03780-f001:**
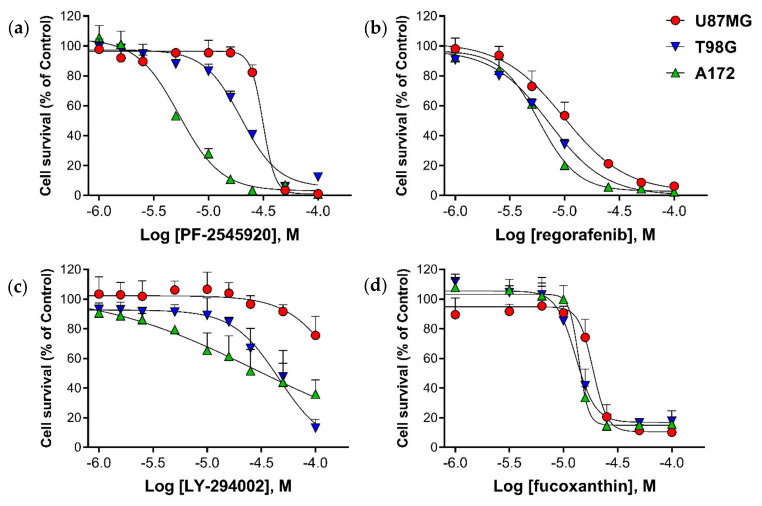
Survival of the three human GBM cell lines U87MG, A172, and T98G, treated with (**a**) Mardepodect (PF-02545920); (**b**) Regorafenib; (**c**) LY-294002; (**d**) Fucoxanthin. Data are presented as the mean ± standard error of the mean (SEM) (*n* = 6–10).

**Figure 2 cancers-13-03780-f002:**
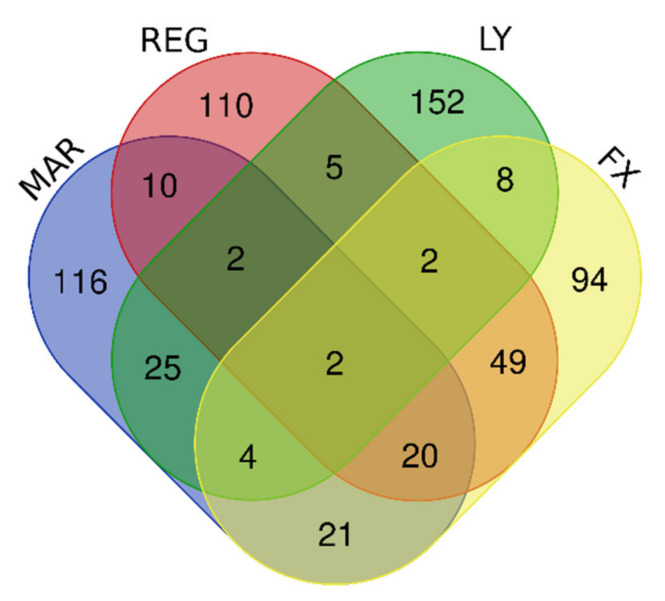
Four-way Venn diagram showing an analysis of the 200 genes with most elevated expression levels in U87MG cells treated with Mardepodect (MAR), Regorafenib (REG), LY-294002 (LY), and Fucoxanthin (FX).

**Figure 3 cancers-13-03780-f003:**
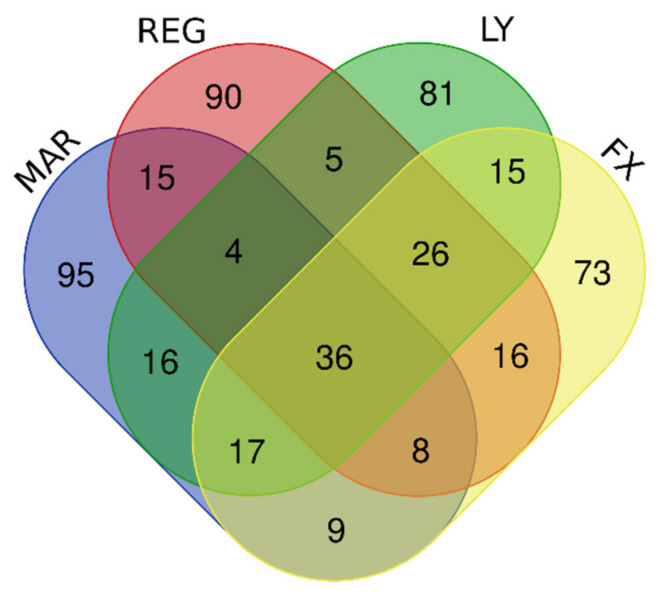
Four-way Venn diagram showing an analysis of the 200 genes with most lowered expression levels in U87MG cells treated with Mardepodect (MAR), Regorafenib (REG), LY-294002 (LY), and Fucoxanthin (FX).

**Figure 4 cancers-13-03780-f004:**
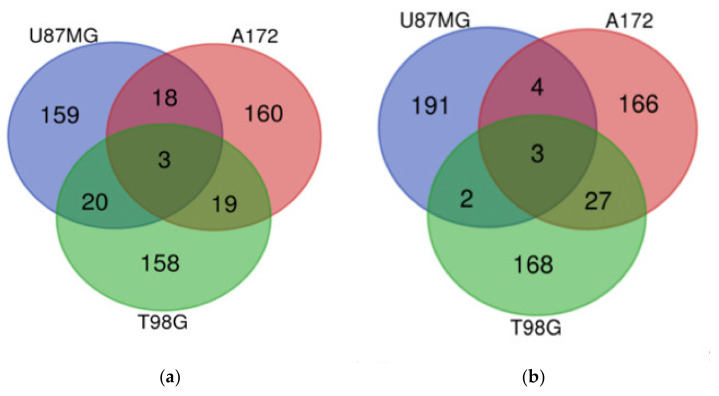
Venn diagrams showing the partitioning of the 200 genes most significantly modulated by Mardepodect in U87MG, A172, and T98G cells; (**a**) upregulated genes; (**b**) downregulated genes.

**Figure 5 cancers-13-03780-f005:**
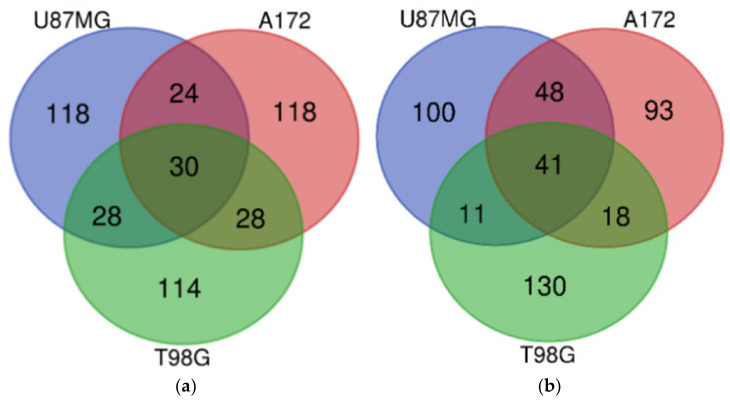
Venn diagrams showing the 200 most significantly modulated genes in Regorafenib-treated U87MG, A172, and T98G cells; (**a**) elevated genes; (**b**) downregulated genes.

**Figure 6 cancers-13-03780-f006:**
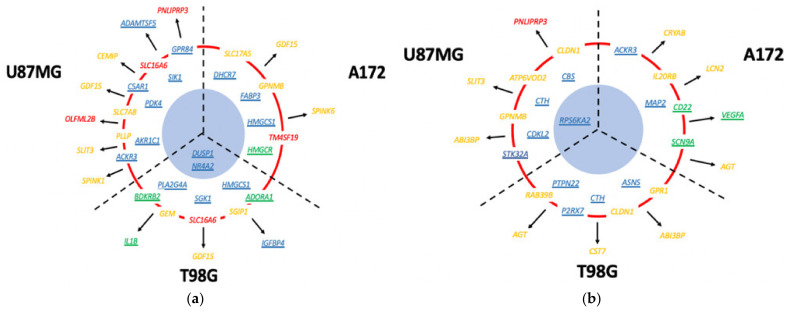
Summary of the most highly elevated drug induced genes encoding potential drug targets, secretory proteins and cell surface antigens expressed within the Top 25 gene set in the three GBM cell lines. (**a**) Mardepodect-treated cells; (**b**) Regorafenib-treated cells. Note the clear differences between the drug modulated phenotypes, both between GBM cell types and individual drug treatments. Targets underlined in green have existing FDA approved drugs; those underlined in blue have chemical leads but no approved drugs; those in yellow have only biological rationales, while those in red remain relatively unexplored.

**Figure 7 cancers-13-03780-f007:**
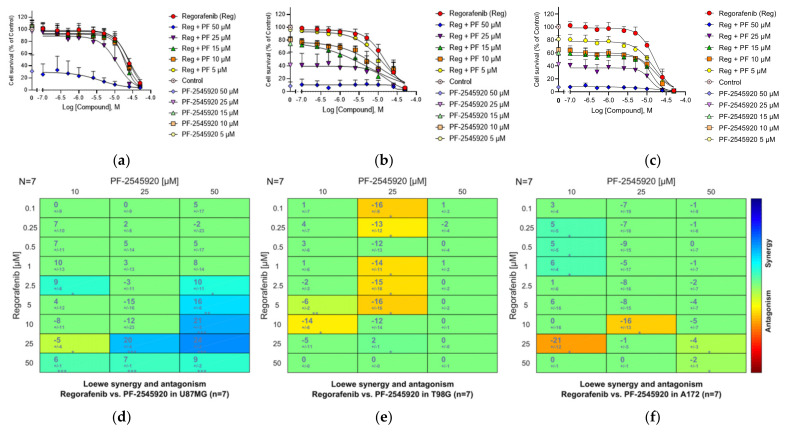
Anti-proliferative effect of combining Mardepodect and Regorafenib; (**a**) U87MG; (**b**) A172; (**c**) T98G cells. Data are presented as the mean ± standard error of the mean (SEM) (*n* = 7); (**d**–**f**), corresponding quantitative analyses of the same data using the Loewe method. Synergy is seen in U87MG cells, but antagonism in both T98G and A172 cells. Note: asterisks indicate the significance of synergy scores obtained following a one-sample *t*-test (* *p* < 0.05; ** *p* < 0.001; *** *p* < 0.0001).

**Figure 8 cancers-13-03780-f008:**
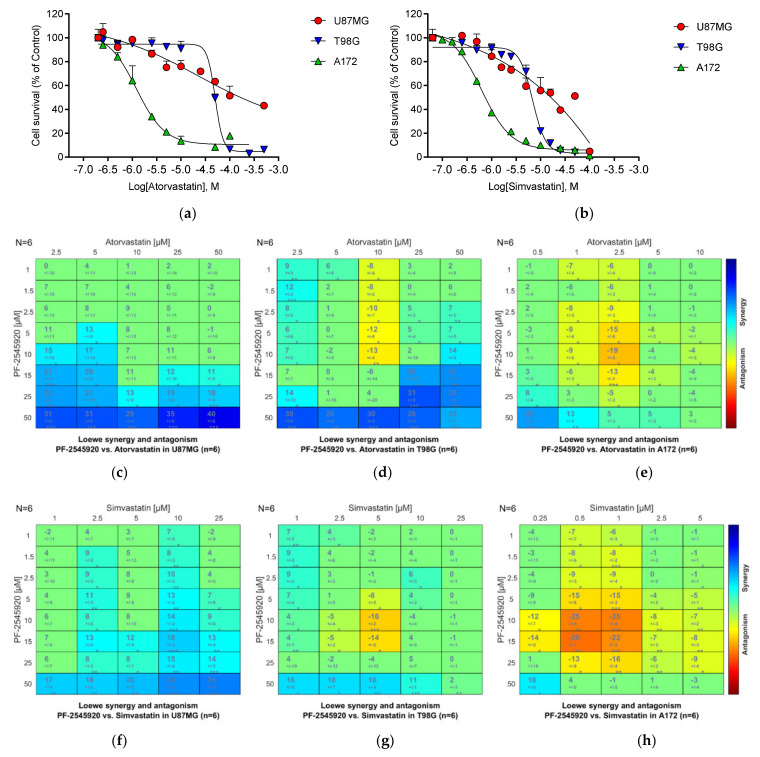
Anti-proliferative effects of HMGCoA reductase inhibitors on U87MG, A172 and T98G cells. (**a**) Atorvastatin; (**b**) Simvastatin. Data are presented as the mean ± standard error of the mean (SEM) (*n* = 3–6). Panels (**c**–**e**) show Loewe synergy plots for the combination of Atorvastatin and PF-2545920 in U87MG, T98G, and A172 glioblastoma cells, respectively. Synergy (in blue) was observed in U87MG and T98G cells at higher Mardepodect concentrations (with a maximal synergy score of 40), but signs of antagonism (in orange) were seen in A172 cells (synergy score of −25). Panels (**f**–**h**) show Loewe synergy plots for the combination of Simvastatin and PF-2545920 in U87MG, T98G, and A172 glioblastoma cells, respectively. Again, some synergy was observed in U87MG and T98G cells, with scores ranging from 16 to 25, with the combinations showing definite antagonism in A172 cells (with a score of −25). Note: asterisks indicate the significance of synergy scores obtained following a one-sample *t*-test (* *p* < 0.05; ** *p* < 0.001; *** *p* < 0.0001).

**Figure 9 cancers-13-03780-f009:**
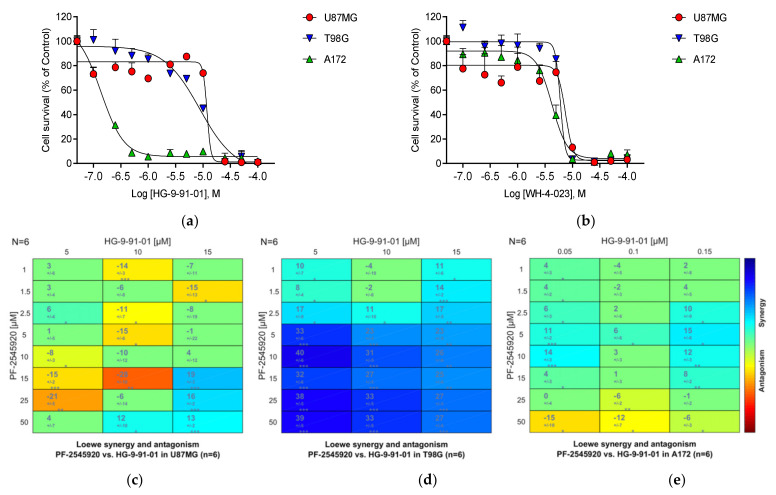
Anti-proliferative effects of SIK1 inhibitors on U87MG, A172 and T98G cells. (**a**) HG-9-91-01; (**b**) WH-4-023. Data are presented as the mean ± standard error of the mean (SEM) (*n* = 3–6). Panels (**c**–**e**) show Loewe synergy plots for the combination of HG-9-91-01 and PF-2545920 in U87MG, T98G, and A172 glioblastoma cells, respectively. T98G cells showed pronounced synergy. Note: Asterisks indicate the significance of synergy scores obtained following a one-sample *t*-test (* *p* < 0.05; ** *p* < 0.001; *** *p* < 0.0001).

**Figure 10 cancers-13-03780-f010:**
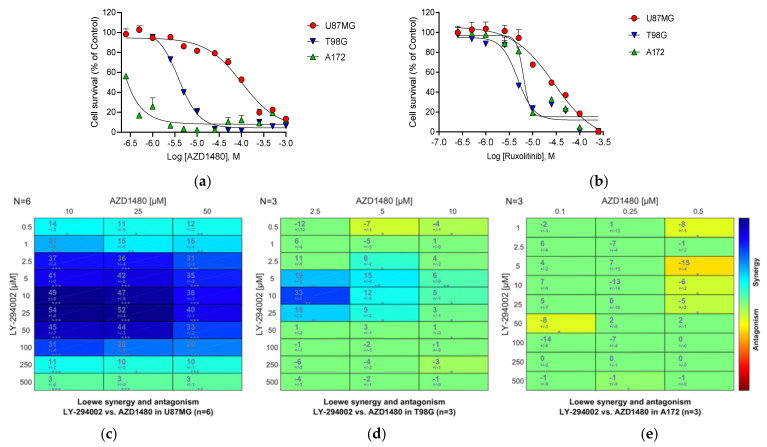
Anti-proliferative effects of JAK2 inhibitors on U87MG, A172 and T98G cells. (**a**) AZD1480; (**b**) Ruxolitinib. Data are presented as the mean ± standard error of the mean (SEM) (*n* = 3–6). Panels (**c**–**e**) show Loewe synergy plots for the combination of AZD1480 and LY-294002 in U87MG, T98G, and A172 glioblastoma cells, respectively. U87MG cells showed pronounced synergy for this combination, with synergy scores rising to >50. Note: Asterisks indicate the significance of synergy scores obtained following a one-sample *t*-test (* *p* < 0.05; ** *p* < 0.001; *** *p* < 0.0001).

**Figure 11 cancers-13-03780-f011:**
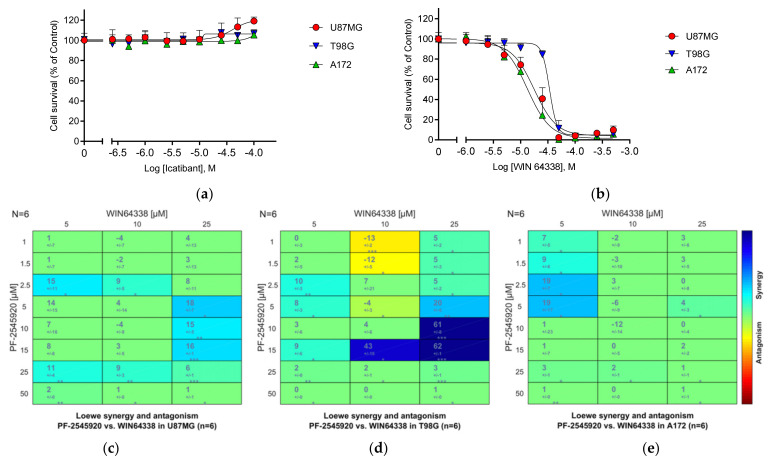
Anti-proliferative effects of bradykinin B2 antagonists on U87MG, A172 and T98G cells. (**a**) Icatibant; (**b**) WIN 64338. Data are presented as the mean ± standard error of the mean (SEM) (*n* = 6). Panels (**c**–**e**) show Loewe synergy plots for the combination of Mardepodect (PF-2545920) and WIN64338 in U87MG, T98G and A172 glioblastoma cells, respectively. T98G cells showed pronounced synergy, with synergy scores rising to 62. Note: Asterisks indicate the significance of synergy scores obtained following a one-sample *t*-test (* *p* < 0.05; ** *p* < 0.001; *** *p* < 0.0001).

**Table 1 cancers-13-03780-t001:** Genes, partitioned between the drug treatments, based on sets of 200 genes with most elevated expression levels in U87MG cells (shown in Figure 2).

Gene Group	Gene Number	Gene Names
Upregulated by Mardepodect, Regorafenib, LY-294002 and Fucoxanthin	2	*PNLIPRP3*, *FAM49A*
Upregulated by Mardepodect and Regorafenib	34	*PNLIPRP3*, *FAM49A*, *PFKFB2*, *WDR78*, *GDF15*, *HMOX1*, *MSC*, *TRIB3*, *GPNMB*, *ERICH2*, *CRYM*, *SLC22A15*, *NUPR1*, *LURAP1L*, *ATP6V0D2*, *CLEC2D*, *GCNT3*, *SLIT3*, *IDH1*, *CTH*, *TM4SF19*, *RFTN2*, *KCP*, *RPS6KA2*, *KIF26B*, *UNC5B*, *PLK2*, *PLXDC2*, *FLYWCH1*, *THBS2*, *PPARGC1A*, *PLEKHF1*, *SLFN5*, *HECW1*
Upregulated by the two multi-kinase inhibitors Regorafenib and LY-294002	11	*ADARB1*, *SOD2*, *TTLL1*, *RSPO3*, *PPIL6*, *GPCPD1*, *H1F0*, *PFKFB2*, *WDR78*, *PNLIPRP3*, *FAM49A*
Upregulated by Mardepodect only, not by Regorafenib, LY-294002 or Fucoxanthin	116	Gene names are found in Appendix A
Upregulated by Regorafenib only, not by Mardepodect, LY-294002 or Fucoxanthin	110	Gene names are found in Appendix A

**Table 2 cancers-13-03780-t002:** Genes, partitioned between the drug treatments, based on sets of 200 genes with most elevated expression levels in U87MG cells (shown in Figure 3).

Gene Group	Gene Number	Gene Names
Downregulated by Mardepodect, Regorafenib, LY-294002, and Fucoxanthin	36	*KIAA1524*, *ESCO2*, *E2F8*, *HIST1H1B*, *LMNB1*, *HIST1H2BB*, *CDCA3*, *HIST2H3A*, *HIST1H2BM*, *TCF19*, *FBXO5*, *HIST1H3B*, *TYMS*, *DNA2*, *ORC1*, *HIST1H2BI*, *FAM111B*, *RRM2*, *ZWINT*, *HIST1H3A*, *ASF1B*, *HIST1H2BH*, *GPR19*, *HELLS*, *PLK4*, *HIST1H2AG*, *RAD54L*, *CDC45*, *HIST1H3F*, *HIST1H2AI*, *SPC25*, *KIFC1*, *KIF15*, *GINS2*, *UBE2T*, *HJURP*
Downregulated by Mardepodect and Regorafenib	63	*ARL14EPL*, *HIST1H4D*, *PBK*, *HIST1H2AB*, *RFC3*, *ATAD2*, *BARD1*, *KIF20A*, *MCM7*, *KIF11*, *HIST1H2BJ*, *TRMU*, *MKI67*, *CENPE*, *ASPM*, *SPAG5*, *KIF4A ANGPTL4*, *ANLN*, *TACC3*, *HIST2H4A*, *HIST2H4B*, *CPA4*, *PLEKHG4B*, *H2AFX*, *GTSE1*, *NCAPG*, *KIAA1524*, *ESCO2*, *E2F8*, *HIST1H1B*, *LMNB1*, *HIST1H2BB*, *CDCA3*, *HIST2H3A*, *HIST1H2BM*, *TCF19*, *FBXO5*, *HIST1H3B*, *TYMS*, *DNA2*, *ORC1*, *HIST1H2BI*, *FAM111B*, *RRM2*, *ZWINT*, *HIST1H3A*, *ASF1B*, *HIST1H2BH*, *GPR19*, *HELLS*, *PLK4*, *HIST1H2AG*, *RAD54L*, *CDC45*, *HIST1H3F*, *HIST1H2AI*, *SPC25*, *KIFC1*, *KIF15*, *GINS2*, *UBE2T*, *HJURP*
Downregulated by Mardepodect only, not Regorafenib, LY-294002, or Fucoxanthin	95	Gene names are found in Appendix A
Downregulated by Regorafenib only, not Mardepodect, LY-294002, or Fucoxanthin	90	Gene names are found in Appendix A

**Table 3 cancers-13-03780-t003:** Genes modulated by Mardepodect, shared between U87MG, A172, and T98G cells.

Sharing Groups	Upregulated	Downregulated
U87MG, A172, and T98G	*GDF15*, *DUSP1*, *SIK1*	*CPA4*, *FAM111B*, *CCL2*
U87MG and A172	*HMOX1*, *SLC11A2*, *GPNMB*, *GPR183*, *UAP1L1*, *PLEKHO1*, *DUSP4*, *LIPG*, *NUPR1*, *PPARGC1A*, *LURAP1L*, *AK5*, *FAM49A*, *RRAGD*, *RRAGC*, *TM4SF19*, *FBXO32*, *RFTN2*	*E2F8*, *TNFRSF11B*, *PI3*, *TXNIP*
U87MG and T98G	*NR4A2*, *CD55*, *HES1*, *SLC16A6*, *S1PR1*, *NR4A1*, *C8orf4*, *CEMIP*, *DNAJB9*, *TNFAIP6*, *SNAI1*, *SGK1*, *ITGB3*, *RND3*, *TRIB1*, *GRAMD4*, *NR4A3*, *ZCWPW2*, *IL6*, *RPS6KA2*	*CD84*, *HIST1H2BM*
A172 and T98G	*HMGCR*, *DDIT4*, *FASN*, *CLCN5*, *SLC2A3*, *JAKMIP2*, *IDI1*, *AGT*, *CYP51A1*, *HLA-DMA*, *SCD*, *FDFT1*, *MVD*, *HMGCS1*, *ST3GAL5*, *DHCR7*, *ZBED8*, *RELL2*, *INSIG1*	*EDN1*, *IRF1*, *PLXNA2*, *TRIM22*, *SERTAD4*, *TNFRSF9*, *TNFSF10*, *RARRES3*, *LGALS9*, *CCNE2*, *KRT18*, *IL7R*, *VCAM1*, *TNFAIP2*, *ENC1*, *RNF150*, *ANKRD1*, *ROR1*, *APOL3*, *CYR61*, *GBP4*, *CTGF*, *PRDM1*, *ALPK2*, *LYPD1*, *BIRC3*, *IL2RG*

**Table 4 cancers-13-03780-t004:** Genes modulated by Regorafenib, shared between U87MG, A172, and T98G cells.

Cell Line Groups	Upregulated Genes	Downregulated Genes
U87MG, A172, and T98G	*TUBE1*, *GDF15*, *TRIB3*, *PTPDC1*, *WARS*, *ERICH2*, *SLC22A18*, *SLC6A9*, *CD22*, *ATF3*, *FAM49A*, *CBS*, *SLFN5*, *TMEM159*, *DDIT3*, *PSAT1*, *IL20RB*, *SOHLH2*, *TTLL1*, *PCK2*, *P2RX7*, *ASNS*, *NUPR1*, *DFNA5*, *AARS*, *CCDC169*, *GTPBP2*, *PPIL6*, *RAB39B*, *KCNH1*	*IL7R*, *ESCO2*, *E2F8*, *MCM3*, *CLSPN*, *DTL*, *HIST1H1B*, *LMNB1*, *PCNA*, *EXO1*, *GINS1*, *MCM6*, *ATAD2*, *BARD1*, *HIST1H2BM*, *SERTAD4*, *MCM10*, *FBXO5*, *POLE2*, *TYMS*, *DNA2*, *MCM5*, *F3*, *ORC1*, *UHRF1*, *FAM111B*, *RRM2*, *HIST1H3A*, *ATAD5*, *HELLS*, *E2F1*, *H2AFX*, *CCNE2*, *SPC25*, *MCM2*, *MCM4*, *FANCB*, *GINS2*, *WDR76*, *HIST1H2AB*, *CDC25A*
U87MG and A172	*ESRP1*, *PKD1L2*, *HMOX1*, *TBL1X*, *KCNT2*, *MSC*, *LURAP1L*, *ANK2*, *UNC5B*, *GPNMB*, *STK32A*, *PHGDH*, *IDH1*, *PIP5KL1*, *THBS4*, *PLPPR4*, *SLC43A1*, *HKDC1*, *TPK1*, *TM4SF19*, *MOCOS*, *PTPN13*, *SCN9A*, *CLIP4*	*TMPO*, *KIAA1524*, *MKI67*, *TGFBR2*, *KIF20B*, *ZGRF1*, *RAD51*, *ASPM*, *LDLR*, *SPAG5*, *RFC3*, *DUSP6*, *CDCA3*, *HIST2H3A*, *LRR1*, *CENPI*, *BRIP1*, *TACC3*, *TCF19*, *SGOL2*, *STIL*, *MCM7*, *CASC5*, *HIST1H3B*, *STARD13*, *KIF11*, *ZWINT*, *ASF1B*, *FEN1*, *HIST1H2BO*, *PLK4*, *RAD54L*, *ZNF367*, *CDC45*, *NCAPD2*, *POLQ*, *PBK*, *NCAPG*, *CDC6*, *HIST1H2BJ*, *POLA2*, *KIFC1*, *ARL6IP6*, *CDCA5*, *UBE2T*, *LIN9*, *HJURP*, *XRCC2*
U87MG and T98G	*TSPAN1*, *PRELID3A*, *DUS4L*, *PPARGC1A*, *TSLP*, *UHRF1BP1*, *STAT2*, *CCDC113*, *TUFT1*, *RCAN1*, *GADD45A*, *SH3BGR*, *CLDN1*, *C6orf48*, *GARNL3*, *TNFRSF9*, *ABI3BP*, *CTH*, *DDR2*, *SLC22A15*, *CCPG1*, *GPR1*, *CCNB1IP1*, *DMGDH*, *GPCPD1*, *ERN1*, *CYP2R1*, *ACAD11*	*STC1*, *MEST*, *CCNF*, *EGLN3*, *SPRY1*, *HIST1H4D*, *HIST1H2BI*, *FAM20C*, *HIST1H2BH*, *EGR1*, *CDK2*
A172 and T98G	*GRB10*, *FYN*, *PCDH1*, *PPP1R3B*, *HOXB9*, *FAM129A*, *SYCP2L*, *SYT14*, *SEL1L3*, *S1PR1*, *SLC1A4*, *THBS3*, *VEGFA*, *OSBPL6*, *ULBP1*, *ARHGEF2*, *SESN2*, *AGT*, *DTNA*, *MAP2*, *CHAC1*, *C10orf107*, *LCA5L*, *CREB5*, *STEAP1*, *CYP4V2*, *ADGRG1*, *AFF3*	*EDN1*, *MMP13*, *HIST1H3H*, *MYCBP*, *HIST1H4L*, *GMNN*, *CCND1*, *CENPW*, *TNFRSF11B*, *HIST1H4A*, *MIS18BP1*, *CCL2*, *TRIB2*, *CDC7*, *PRDM1*, *CSNK1G1*, *MYB*, *HIST1H2BF*

## Data Availability

All data used in this publication are reported within it.

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
