# Peer review of "Transcriptomics-Based Phenotypic Screening Supports Drug Discovery in Human Glioblastoma Cells"

_cancers, 2021, doi:10.3390/cancers13153780_

Round 1

Reviewer 1 Report

Dear Authors,

Shapovalov et al, paper titled ‘The Drug Discovery Landscape of Human Glioblastoma Cells’ is a research article addressing an extremely important gap in the field concerning the lack of effective treatments for GMB patients. The authors use genome-wide drug-induced gene expression (DIGEX) technique to gain insights into GMB expression changes in response to therapy and infer what could be novel therapeutic targets. Following an in-depth analysis of the upregulated and downregulated genes, the audience of the paper can see a few attempts to validate the author’s DIGEX findings.

The manuscript has a potential to become an excellent paper, however there are a few key elements that need to be addressed to improve this study:

  1. The authors indicate throughout the manuscript that the 3 cell lines under investigation respond differently to Regorafenib, Mardepodect, LY-294002 or Fucoxanthin and that U87MG, A172 and T98G have been completely sequenced (line 180-181). The audience of the paper could greatly benefit from an in-depth analysis of the differences between these cell lines at the genomic and transcriptional level.
    Do these cell lines have mutations in any proteins that would impact their response to Regorafenib, Mardepodect, LY-294002, Fucoxanthin or any of the other inhibitors mentioned in the paper? Likewise, what are the expression levels of the ‘top 200 genes’ from every data set before Regorafenib, Mardepodect, LY-294002 or Fucoxanthin treatments? For instance, Table 3 indicates there are only 3 upregulated genes shared between the 3 cell lines upon Mardepodect treatment, but perhaps the 18 upregulated genes shared between U87MG and A172 were already highly expressed at baseline in T98G and that’s the only reason why the upregulation upon treatment wasn’t observed. Conversely, perhaps no downregulation of certain genes was seen because the levels at baseline were already low. The authors should already have the necessary data to address these questions and perform the analysis.
  2. In section 3.4 lines 210-215 the authors indicate their expression analysis was conducted on cells after 24h treatment, while the proliferation analyses after 72h. If the results of the study were to be used to identify a druggable target, the patients would have been treated for an extended amount of time. It would be important to verify (for at least one of the drugs/cell lines) if the expression changes after 24h of treatment persist over prolonged exposure e.g. 1 week, 1 month. Given the cell plasticity, would they develop resistance to treatment and as a result revert to their previous expression pattern or acquire even more changes?
  3. Except for HMGCoA Reductase inhibitors (Lines 768-769), the outcome regarding the combination treatment does not immediately follow the paragraph presenting data on the monotherapy for any given putative GMB target. Instead, like in the case of SIK1, the audience of the paper reads the first half of the story in paragraph 3.22 and the second half in paragraph 3.27.3, separated by the results of multiple other targets. The separation between monotherapies and drug combinations seems artificial. The readability of the paper would be greatly improved if all experiments concerning any given target were presented as a whole.
  4. In figures 10, 11 and 12 the authors present interesting results concerning susceptibility of the GMB cell lines to A1R antagonist SLV320, non-peptide B2R antagonist WIN 64338, and pancreatic lipase (PNLIP) inhibitor Orlistat (respectively). However, unlike in case of HMGCoA Reductase, SIK1 or JAK2 inhibitors, the combination studies aren’t performed. The quality of the paper will greatly improve when this weakness is addressed, and studies conducted.
  5. In paragraph 3.26 discussing drug combinations, the authors attempt to observe synergy between treatments, but no equations or proper analysis can be found. Line 954 indicates there is a ‘modest’ synergy between Mardepodect and Regorafenib in the three GBM cell lines, however Figure 13 legend (line 960) directly contradicts this statement indicating there is ‘very little’ synergy. Claims of ‘little’ or ‘strong’ synergy appear throughout the manuscript but these terms are not defined or supported by quantification.

Lesser points that would further improve the paper:

  1. Throughout the manuscript: please provide % increase or fold increase for the discussed genes. 
  2. Line 71: FDA registers medical devices (some need approval). All drugs need FDA approval (not registration). Please rephrase.
  3. Lines 203-206: Presenting IC50s as exponents; especially when fractions are involved, is not intuitive and will impair the audience’s ability to compare the results of this study with other publications which express drug IC50s as rational numbers.
  4. Line 259: Please substitute ‘close homology’ with appropriate percentage and specify whether the homology concerns the entire protein or the catalytic domain alone.
  5. Line 314: Please spell out ‘ER’
  6. Lines 375-378: please provide literature examples of ‘controlled program of transcriptional rebalancing’
  7. Lines 395-400: perhaps better to include as a part of Discussion?
  8. Lines 520-522: speculative until tested, unless the authors intend to add this study to the paper, perhaps best to move it to the Discussion?
  9. Lines 603-606: what is so special about U87MG cells that could explain this? How would the U87MG phenotype look like in patients?
  10. Lines 768-769: please show the data as supplementary material.
  11. Lines 784-785: Table 3 indicates SIK1 is upregulated, not Figure 4.
  12. Lines 784-785 and 801-802: These 2 sentences are directly contradicting each other; one states all 3 GMB cell lines highly upregulate SIK1 upon Mardepodect treatment; the other states SIK1 is being highly induced in U87MG only.
  13. Lines 855-856: which cell lines was the upregulation seen in and upon response to which drug(s)?
  14. Supplementary S2+: please provide a legend and define what do the gray lines indicate? Are these direct or indirect interactions? One directional or bidirectional etc? 

Reviewer 2 Report

Shapovalov et al use DIGEX in three human GBM cell lines to determine the transcriptomic response to the treatment with two drugs of interest in vitro. The RNA-seq data were reproducible and there was some overlap as well as differences. between the response of the cell lines. The authors furthermore used their transcriptomic data to identify potential drug combinations. The paper is written clearly and the figures are presented appropriately. The main concerns with the paper are:

  1. A large number of genes will necessarily be up or downregulated in response to drug treatment; however, many of these genes are not likely to be drug candidates because they have likely have no functional role in these cells' proliferation or drug resistance but are maybe a consequence of the perturbation induced. Much of the results' section is discussion of genes up or downregulated without further experiments exploring their function in these lines; these points should either be moved to Discussion where relevant or shortened. Indeed the authors even point out that true mediators of cell growth may be 'buried' in the RNA-seq data. Further reference and discussion should be made to in vivo functional studies using transposons or CRISPR whereby the targets are directly identified through genetic mutations generating a functional response (such approaches provide a stronger link between causation and effects - for example, Chow et al Nat Neurosci 2017, CRISPR; Noorani et al 2020 Genome Biol , PiggyBac mutagenesis; also further discussion in Chow et al 2018, Cancer CRISPR screens in vivo; Noorani et al 2020 Genome Biol, CRISPR and transposon screens in vivo).
  2. The drug combinations explored need further validation - for example gene silencing / KO could be performed to confirm the drug target is the gene identified. 
  3. Consideration should be given to validation of targets with patient tissue samples; these cell lines, although established, have been passaged a great deal and may have significant divergence from patient GBM or mouse model biology. These points must be discussed at least. 
  4. In line with point 1, the title should be altered appropriately - it is too far reaching currently to state this as the drug discovery landscape given the points regarding the relevance of many genes altered in expression. 

Round 2

Reviewer 2 Report

The authors have addressed my concerns